# Spatial Distribution of Displaced Population Estimated Using Mobile Phone Data to Support Disaster Response Activities

Silvino Pedro Cumbane [1,2,*] and Győző Gidófalvi [1]

1 Division of Geoinformatics, Department of Urban Planning and Environment,
KTH Royal Institute of Technology, Teknikringen 10A, SE-114 28 Stockholm, Sweden; gyozo@kth.se

2 Division of Geographic Information Sciences, Department of Mathematics and Informatics,
Eduardo Mondlane University, Julius Nyerere Street, Maputo 3453, Mozambique

* Correspondence: silvino@kth.se; Tel.: +46-728781208 or +258-849226566

**Abstract:** Under normal circumstances, people's homes and work locations are given by their addresses, and this information is used to create a disaster management plan in which there are instructions to individuals on how to evacuate. However, when a disaster strikes, some shelters are destroyed, or in some cases, distance from affected areas to the closest shelter is not reasonable, or people have no possibility to act rationally as a natural response to physical danger, and hence, the evacuation plan is not followed. In each of these situations, people tend to find alternative places to stay, and the evacuees in shelters do not represent the total number of the displaced population. Knowing the spatial distribution of total displaced people (including people in shelters and other places) is very important for the success of the response activities which, among other measures, aims to provide for the basic humanitarian needs of affected people. Traditional methods of people displacement estimation are based on population surveys in the shelters. However, conducting a survey is infeasible to perform at scale and provides low coverage, i.e., can only cover the numbers for the population that are at the shelters, and the information cannot be delivered in a timely fashion. Therefore, in this research, anonymized mobile Call Detail Records (CDRs) are proposed as a source of information to infer the spatial distribution of the displaced population by analyzing the variation of home cell-tower for each anonymized mobile phone subscriber before and after a disaster. The effectiveness of the proposed method is evaluated using remote-sensing-based building damage assessment data and Displacement Tracking Matrix (DTM) from an individual's questionnaire survey conducted after a severe cyclone in Beira city, central Mozambique, in March 2019. The results show an encouraging correlation coefficient (over 70%) between the number of arrivals in each neighborhood estimated using CDRs and from DTM. In addition to this, CDRs derive spatial distribution of displaced populations with high coverage of people, i.e., including not only people in the shelter but everyone who used a mobile phone before and after the disaster. Moreover, results suggest that if CDRs data are available right after a disaster, population displacement can be estimated, and this information can be used for response activities and hence contribute to reducing waterborne diseases (e.g., diarrheal disease) and diseases associated with crowding (e.g., acute respiratory infections) in shelters and host communities.

**Keywords:** disaster response; mobile Call Detail Records (CDRs); displaced population

## 1. Introduction

Evacuation plans are very important tools to guide people in different emergency situations on how to evacuate. However, when a disaster occurs, even though there are predefined shelters such as schools, communal halls, libraries, and other buildings, in many cases, people have an instinctive feeling as to the direction of safety [1]. They tend to move away from danger and towards destinations perceived as safe [2]. In addition to that, the evacuation behaviors also depend on whether all shelters in the surrounding area

can provide refuge for people at a reasonable evacuation distance and time immediately after a disaster [3,4]. The human behavior in different emergency contexts is explained by social and psychological theories and associated underlying cognitive and behavioral processes [5–7]. Moreover, in some cases, depending on the intensity of a disaster, shelters may suffer damage themselves or fail to provide the safety and security needed, forcing people to move to places other than shelters [8]. Therefore, when a disaster has struck, people in shelters do not represent 100% of the displaced population, which brings a challenge to government and humanitarian aid organizations when it comes to allocating resources for disaster response.

Knowing the spatial distribution of the displaced population (including people that moved to shelters and other places) is crucial for rapid and effective disaster response, which among other things aims to meet humanitarian needs such as food, clothing, public health, and safety [9,10]. Traditional methods of estimation of people displacement are based on population survey. For example, Hori et al. [11] used the 2006 Louisiana Health and Population Survey (LHPS) to describe three distinct dimensions of displacement dynamics: in-migration, out-migration, and intra-parish movement in southern Louisiana after hurricanes Katrina and Rita, on 29 August 2005 and 24 September 2005, respectively. Gray et al. [12] used large-scale survey data collected from respondents living in coastal areas of Indonesia before and after the 2004 Indian Ocean tsunami. Survey data were combined with satellite imagery and multivariate statistical analyses to map vulnerability to post-tsunami displacement across the provinces of Aceh and North Sumatra. In addition to that, the survey data were used to compare patterns of migration after the tsunami between damaged areas and areas not directly affected by the tsunami. Recently, governments and other humanitarian aid organizations have been using information about the spatial distribution of people in shelters provided by the Displacement Tracking Matrix (DTM) after surveying the affected populations [13].

However, conducting a survey is time-consuming and infeasible to perform at scale and provides low coverage, i.e., can only cover the numbers for the population that are at the shelters. In addition, the information cannot be delivered in a timely fashion. For example, cyclone Idai struck Beira city on 14 March 2019, but the survey by DTM in close coordination with Mozambique's National Institute for Disaster Management (INGC) was conducted between 2 and 13 May 2019, i.e., a month and half after disaster [14]. The result showed that by the time the survey was conducted, some people had started to return to their areas of origin, which means that using this information government and other humanitarian agencies could not assist people when they needed it. Therefore, the spatial distribution of the displaced population in a timely fashion is necessary.

Many approaches have been proposed to model the people's behavior during the evacuation and to estimate the spatial distribution of displaced people after a disaster to support response activities. For instance, Osaragi [15] constructed different models to understand the behavior of individuals attempting to reach home on foot in the wake of a devastating Tohoku-Pacific Ocean Earthquake (2011). Hu et al. [16] used the susceptible–infective-removal (SIR) model to understand the spread of disaster risk perceived among homeless victims and other disaster-affected people while considering the effects of psychological interventions on them. Yabe et al. [17] proposed a novel framework to estimate evacuation hotspots after the Kumamoto earthquake using Global Positioning System (GPS) traces of smartphones collected by Yahoo Japan. The experiment showed promising results. However, the sample data (approximately 1% of the population from all over Japan) were very limited. To overcome sample bias, anonymized mobile Call Detail Records (CDRs) data have been proposed as an alternative source of information to estimate the spatial distribution of the population used as a base to estimate the displacement matrix after a disaster. For example, Kubíček et al. [18] used mobile phone data to model population distribution at a fine spatio-temporal scale in the city of Brno, Czech Republic, and discussed the potential use of the proposed approach within selected emergencies. The analysis was based on the number of people visiting and transiting in each specific area. The results

of the analysis were compared to census data and proved how the proposed method can improve the spatial granularity of the number of visiting and transiting in each specific area. Li et al. [19] used large-scale mobile phone data to estimate fine-grained dynamic population distribution and high-resolution population map in Shanghai city, China, and analyzed spatio-temporal interaction of human movement. Recently, Wang et al. [20] presented a systematic literature review paper showing the different applications of mobile phone data for emergency management. The review was based on the analysis of 65 related articles written between 2014 and 2019 from six electronic databases. Zhang et al. [21] proposed a method for the estimation of finely-grained spatio-temporal human population density distributions using mobile CDRs. Moreover, Balistrocchi et al. [22] discussed the suitability of mobile phone data to derive crowding maps. Through this study, characteristic exposure spatio-temporal patterns and their uncertainties were detected with regard to land cover and calendar period. However, the proposed approach still deserves verification during real-world flood episodes.

Bengtsson et al. [23] used position data of Subscriber Identity Module (SIM) cards from the largest mobile phone company in Haiti (Digicel) to estimate the magnitude and trends of population movements following the Haiti 2010 earthquake and cholera outbreak. The experimental study was conducted in Port-au-Prince and proved that the geographic distribution of population movements from Port-au-Prince corresponded with results from a large retrospective, population-based UN survey. Lu et al. [24] used the same data (from Digicel) to evaluate the predictability of population displacement after the 2010 Haiti earthquake. The authors found that both the travel distances and size of people's movement trajectories grew after the earthquake. In addition to that, the results suggested that the predictability of people's trajectories remained high and even increased slightly during the three months after the earthquake. Another study was conducted by Wilson et al. [25]. In this study, the authors proposed a method for rapid and near real-time assessments of population displacement following the 2015 Nepal earthquake by analyzing the movement of 12 million anonymized mobile phone users. The result showed the evolution of population mobility patterns after the earthquake and the patterns of return to affected areas at a high level of detail.

However, the methods presented by Bengtsson et al. [23] and Wilson et al. [25] rely on defining the daily user's home location as the last place he/she made a call from, which in many cases is not realistic since a person can have their last mobile activity before getting home. Furthermore, even while at home, a mobile activity can be placed in a cell-tower that is far away from the user's home location due to overload in the nearest cell-tower. Moreover, each user is assigned a daily location at the district level based on the location that the corresponding user's home cell-tower is in, without considering that the cell-tower can be shared among different districts. The aim of this research is to estimate the spatial distribution of displaced people after a severe cyclone using anonymized mobile phone data.

Therefore, in this research, were adopted some proven techniques from different studies such as study area tessellation [26], home location estimation [25,27], areal interpolation [28], and origin-destination estimation [27,29] to derive a method for the spatial distribution of displaced population after a disaster. This method contributes in three aspects, namely: (1) user's home location (at cell-tower level) is based on the frequency of usage of mobile phone during the night time; (2) each user is assigned a home location (at the neighborhood level) considering the coverage area of the cell-tower that he/she was found to be living in; (3) to evaluate the accuracy of the proposed method, validation was introduced, which compares the displaced population estimated using CDRs with remote-sensing-based building assessment data and Displacement Tracking Matrix (DTM) from an individual questionnaire survey conducted after a severe cyclone in Beira city, central Mozambique, in March 2019. The proposed method can be used to derive a near-real-time displaced population matrix after a disaster, which in turn can support the response teams

in their activities that, among others, aim to provide for the basic humanitarian needs of affected people.

The remainder of the paper is organized as follows. Section 2 presents the study area and the methods used in this research. Section 3 describes the data used in this research. Section 4 presents experimental results of the proposed method and the validation of the estimated displacement matrix. Finally, Section 5 concludes and presents future directions of the research.

## 2. Study Area and Methods

In this section, first, the methodology is presented by giving an overview (Figure 1). Then, the study area is described. The last part in this section describes the method, step by step, following the overview.

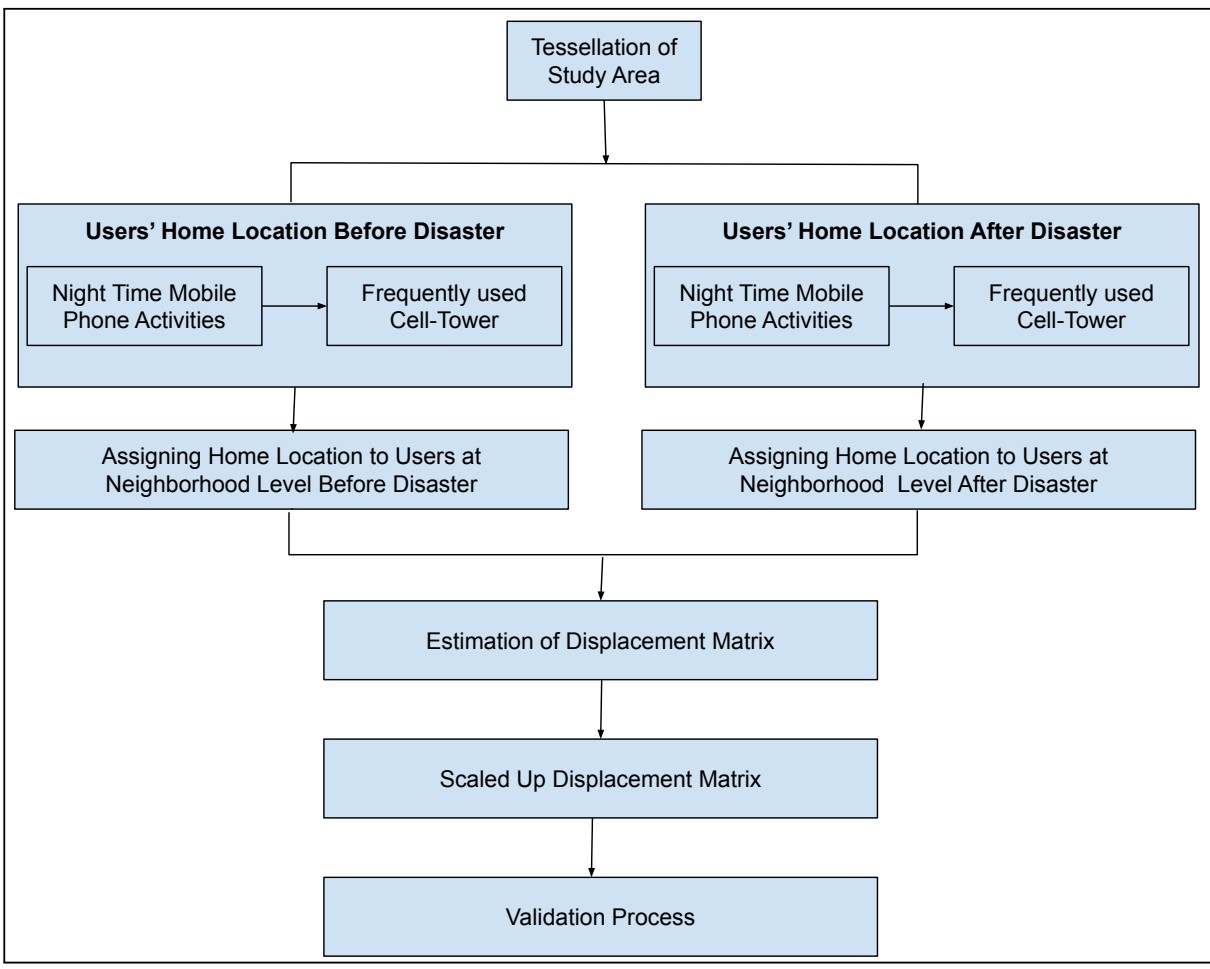

**Figure 1.** Methodology flow chart.

### 2.1. Study Area

The study was conducted in Beira city with an area of 631 km$^2$, located in the central region of Mozambique in Sofala Province, where the Pungwe River meets the Indian Ocean. Beira city consists of 26 neighborhoods [30]. Recently, Chamba has been mentioned as a new neighborhood in Beira city [31,32]. Therefore, in this paper, it is assumed that Beira city has 27 neighbourhoods. Neighborhood is an immediate geographical area surrounding a family's place of residence, bounded by physical features of the environment such as streets, rivers, train tracks, and political divisions [33]. Beira city had a population of 530,604 inhabitants in 2020 and holds the regionally significant Port of Beira, which acts as a gateway for both the central interior portion of the country as well as the land-locked

nation of Zimbabwe, Zambia, and Malawi [34]. The city is located just a few meters above sea level, which makes it vulnerable to climate-related threats. Cyclone Idai hit Beira on 14 March 2019 and proved the vulnerability of the city and if it had come with the higher tide, its devastating effects would have been even worse [35]. Cyclone Idai caused massive flooding, leaving entire communities submerged under 10 m of water, and damaged infrastructure and roads, affecting 3 million people. As of 12 April 2019, the number of houses destroyed was 239,731, of which 112,745 were totally destroyed, 111,202 partially destroyed, and 15,784 flooded [36]. Figure 2 shows the study area and its neighborhood boundaries and the shapefile used to produce it was acquired in [37].

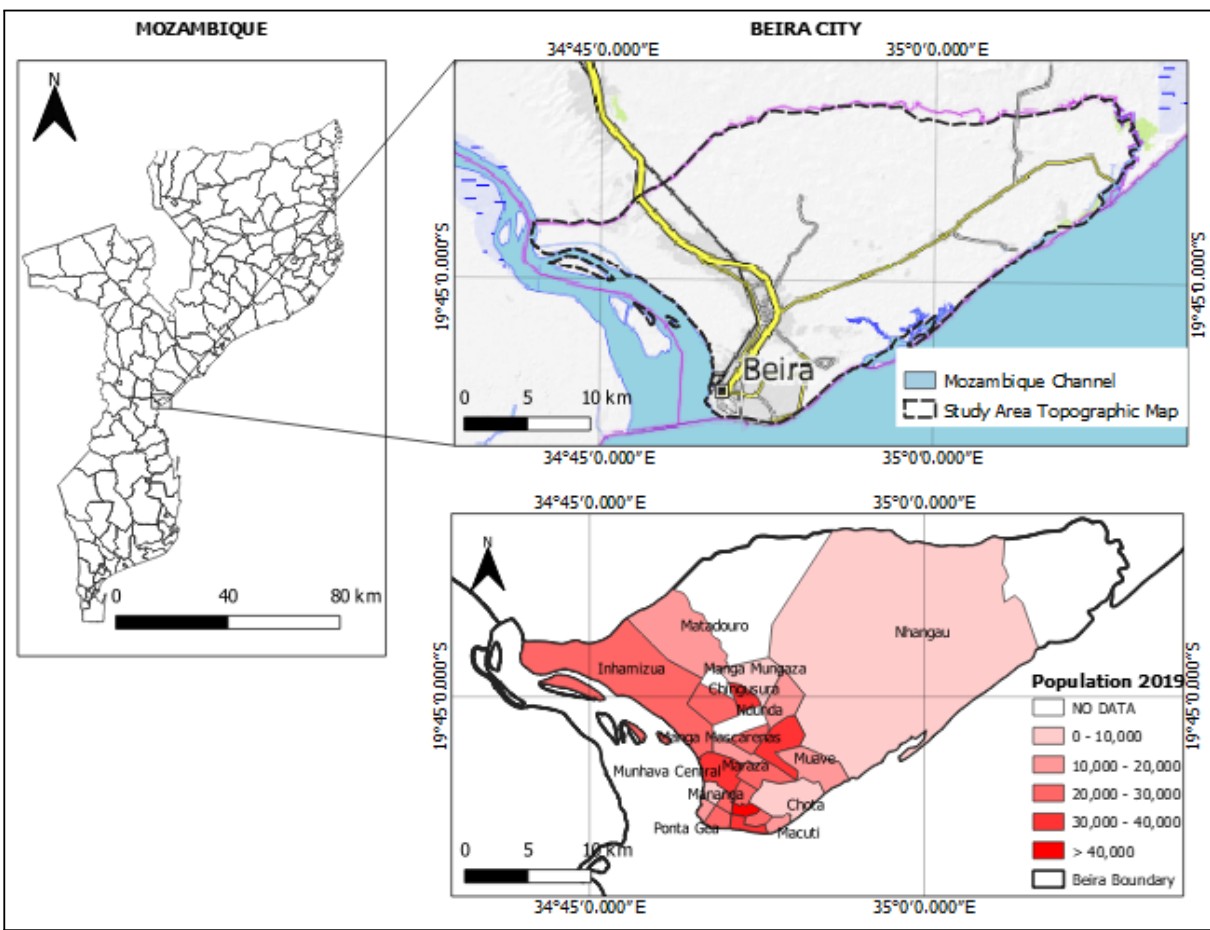

**Figure 2.** Study area—Beira city and its neighborhood boundaries.

## 2.2. Methods

Estimation of the displaced population using anonymized mobile CDRs consists of multiple steps, and some of them were performed using proven techniques from previous research [25,38]. The proposed method involves six main steps, namely (1) Voronoi tessela-tion of the study area, (2) estimation of mobile phone users home location (at cell-tower level) before and after disaster, (3) neighborhood home location assignment before and after disaster, (4) estimation of displaced mobile phone users, (5) scaling up the displaced mobile phone users to actual population flow, and (6) validation process. The data processing was mainly carried out using Python API of Spark processing Framework (PySpark) as suggested by Cumbane and Gidófalvi [39].

### 2.2.1. Voronoi Tesselation of the Study Area

The study area consists of 51 cell-towers spatially distributed according to the popula-tion density; i.e., highly populated areas have a dense distribution of cell-towers. Hence,

the coverage area of each cell-tower decreases in highly populated areas. Using cell-towers locations as centroids, Voronoi polygons of the study area were constructed using principles presented by Okabe et al. [26], as shown in Figure 3.

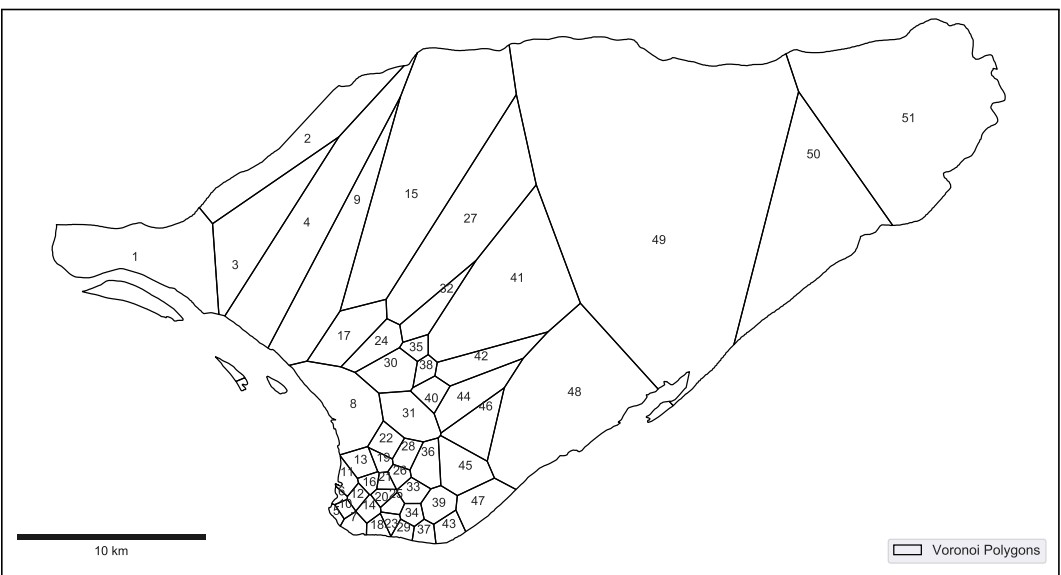

**Figure 3.** Voronoi polygons of the study area.

Each Voronoi polygon approximately represents the mobile phone cell-tower coverage area. Figure 3 clearly shows that the areas of the Voronoi polygons increase from the central city of Beira to the sub-urban areas. The smallest and largest Voronoi polygons have 0.37 km² and 14,544 km² of coverage area, respectively. Furthermore, the mean and median coverage areas are 14.02 km² and 2.17 km², respectively.

2.2.2. Estimation of Mobile Phone Users Home Location before Disaster

The underlying principle in estimating the user's home location is that people tend to stay at home during the night time. Using this principle, a cell-tower is defined as the user's home location if it is the most frequently used cell-tower during the night time for the entire period before the disaster. To select a cell-tower as the user's home location, the following steps have to be considered:

- the daily most frequently used cell-towers during the night time are selected (candidates' home locations),
- candidates' home locations are aggregated for the entire period of analysis (before the disaster),
- the most frequent candidate home location is selected ast the user's home location before the disaster.

To reduce biases in the estimation, only the candidates with a frequency greater or equal to 2 were selected as user's home location before the disaster. This means that a candidate has to have been identified at least twice as the night-time user's home location to be considered their home location before the disaster. Users to whom this condition does not apply were ignored from the analysis.

One important aspect when selecting users' home locations is to define the night time period. Defining a suitable night time period was challenging because no household survey data were found about the study area. However, according to Maloa [40], Maputo (capital of Mozambique) and Beira (study area) have similarities in terms of physic-geographic characteristics and mobility activities. Therefore, in this study, night time is considered to be from 08:00 p.m. to 06:00 a.m. because this time was found to be when most of the people in Great Maputo are at home according to the JICA Household Survey [41].

**Example 1.** *Let us suppose that an user $U$ has activities registered for four days before the disaster. The activities $U$ are associated with the following cell-towers $c_i$ per day $d_j$: $d_1 = \{c_1, c_{21}, c_{21}, c_{21}, c_{44}, c_1, c_1, c_1\}$, $d_2 = \{c_1, c_{21}, c_{21}, c_1, c_1, c_2\}$, $d_3 = \{c_1, c_1, c_{21}, c_{21}, c_{44}, c_1, c_{43}\}$, $d_4 = \{c_2, c_1, c_1, c_{21}, c_{21}, c_1, c_2\}$. Let us suppose that the night-time activities $n$ for each day are: $n_1 = \{c_{44}, c_1, c_1, c_1\}$, $n_2 = \{c_1, c_1, c_2\}$, $n_3 = \{c_{44}, c_1, c_{43}\}$, $n_4 = \{c_1, c_2\}$. The candidate user's home cell-tower for each day $cd_i$ is defined as the most frequent cell-tower during the night time for each day. Therefore, $cd_1 = \{c_1\}$, $cd_2 = \{c_1\}$, $cd_3 = \{c_{43}\}$, $cd_4 = \{c_1\}$. Note that in order to simplify the approach, in those cases where there is no clear candidate ($n_3$ and $n_4$), the cell-tower is randomly selected, as can be seen in $cd_3$ and $cd_4$. Finally, the user's home location is defined as the most frequent candidate for the entire analysis period. Thus, for this particular case, user $U$'s home location (at cell-tower level) is defined as $c_1$ since it has the highest frequency (3) and is greater than 2.*

2.2.3. Estimation of Mobile Phone Users Home Location after Disaster

The principle applied to estimate the mobile phone user's home location (at cell-tower level) before a disaster is applied to extract their home location after a disaster. However, for this estimation, only the users who were identified to be living in the study area before the disaster were considered; i.e., all users who moved to the study area after the disaster were ignored from the analysis. For the experimental purpose, the data from 2 to 8 April were used. Since the users considered after the disaster were those found to be living in the study area before the disaster, a particular user's home location (at cell-tower level) after the disaster was selected as the most frequent cell-tower among candidate cell-towers. Note that in this case, the condition that the frequency should be greater than or equal to 2 is not considered in order to incorporate those users that have only one candidate home location after the disaster in the displacement matrix.

2.2.4. Assigning Home Location to Users before and after Disaster at Neighborhood Level

Users (at cell-tower level) were assigned to neighborhood administrative boundaries using the areal interpolation method proposed by Flowerdew et al. [28]. To achieve this goal first, the Voronoi polygons derived from cell-towers locations were overlapped with the neighborhood administrative boundaries data. Secondly, the intersection areas between Voronoi polygons and neighborhood data and corresponding percentages were calculated. Thirdly, based on the number of users who were found to be living in a given cell-tower area before or after a disaster and the percentage of intersection areas between cell-tower and a particular neighborhood, the corresponding number of users to be assigned to this neighborhood was calculated. Finally, the users were randomly assigned to the neighborhoods.

**Example 2.** *Let us assume that Figure 4 represents the overlapping result between Voronoi polygons (dashed lines) and the neighborhood boundaries (solid lines).*

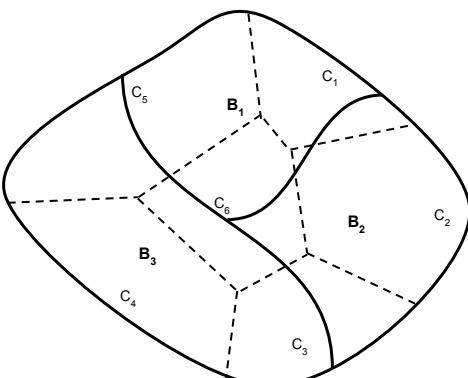

**Figure 4.** Example of overlay between voronoi and neighborhood administrative boundaries.

In Figure 4, $c_1, c_2, c_3, c_4, c_5$, and $c_6$ represent the Voronoi polygons for cell-tower 1, 2, 3, 4, 5, and 6, respectively. $B_1, B_2$, and $B_3$ represent the boundaries of neighborhood 1, 2, and 3, respectively. The task is to assign a user $U_i$ to a particular neighborhood given that he/she was found to be living in a particular cell-tower $c_k$. This task can be easy if the Voronoi polygon is not shared by different neighborhoods ($c_4$), but, it can be complex if the cell-tower coverage area is shared by different neighborhoods ($c_1, c_2, c_3, c_5$, and $c_6$). For the first case (when the Voronoi polygon is not shared by different neighborhoods), all the mobile phone users who were found to be living, for example, in $c_4$ were assigned to neighborhood $B_3$. However, in those cases where a cell-tower is shared among different neighborhoods, the following solution was proposed to assign the mobile phone users to neighborhoods.

Let us use the cell-tower $c_6$ as an example. This cell-tower is shared among all the three neighborhoods, $B_1, B_2$, and $B_3$ and we assume that it has homogeneous areas in terms of inhabitants' density. Similar case from study area is shown in Figure 5.

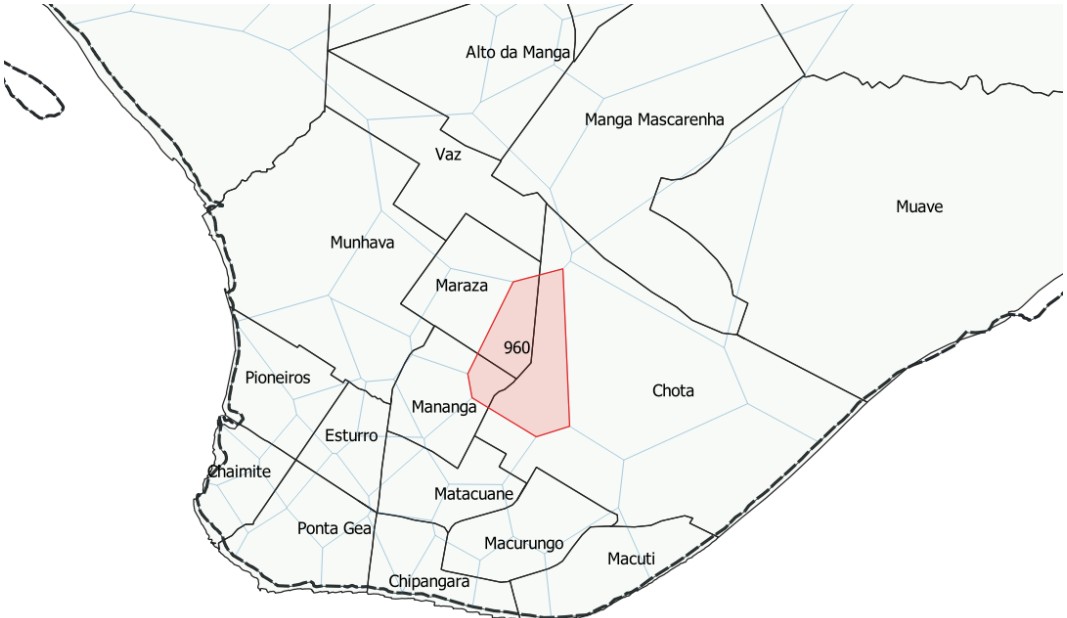

**Figure 5.** Example of overlay between voronoi and neighborhood administrative boundaries from study area.

In Figure 5, the red polygon (cell-tower 960) is shared among three neighborhood, namely Maraza, Mananga, and Chota. In Figure 4, to assign the mobile phone users that were found to be living in this cell-tower, first, the area of Voronoi polygon that falls in each neighborhood is calculated based on the geometry intersection between the Voronoi polygon data and neighborhood administrative boundaries and the corresponding result would be $a_1, a_2, a_3$. The percentages $p_i$ of $a_1, a_2, a_3$ are calculated using Equation (1).

$$p_i = \frac{a_i}{A_t}, \quad i = 1, 2, 3 \tag{1}$$

where:

- $a_i$ is the intersection area between Voronoi polygon data and neighborhood administrative boundaries; and
- $A_t$ represents total area of $c_6$ in the example.

Then, the number of mobile phone users $nu_i$ that have to be assigned in each neighborhood is calculated using Equation (2).

$$nu_i = p_i * tu, \quad i = 1, 2, 3 \tag{2}$$

where:

- $p_i$ represents the percentages of $a_1, a_2, a_3$; and
- $tu$ is the total number of users that were found to be living in the voronoi polygon $c_6$.

Based on the number of mobile phone users $nu_i$ calculated using Equation (2), the corresponding mobile phone users are randomly selected from $c_6$ and assigned to each neighborhood. This process is repeated for all the cell-towers that are shared among different neighborhoods before and after a disaster.

### 2.2.5. Estimation of Displacement Matrix of Mobile Phone Users

After assigning the users to neighborhoods before and after the disaster, the next step is to estimate the number of displaced people. A user $U_i$ is defined as displaced if their home neighborhood before the disaster $B_b$ is different from the home neighborhood after disaster $B_a$, as shown in Equation (3).

$$U_i = \begin{cases} \text{displaced,} & \text{if } B_b \neq B_a \\ \text{non-displaced,} & \text{if } B_b = B_a, \quad i = 1, 2, ..., N \end{cases} \tag{3}$$

In order to estimate the displacement matrix of mobile phone users, first, mobile phone users' home location (at neighborhood level) data before and after the disaster were joined based on the anonymized user's ID. Then, the users were aggregated based on their home location neighborhood ID before and after a disaster, and the mobile phone users' displacement matrix was derived.

### 2.2.6. Scaling up Mobile Phone Users' Displacement Matrix

Usually, governmental and non-governmental institutions require information about total population flows, rather than the subset represented by mobile phone users. Therefore, assuming that the data used in this study are representative of the population in the study area, the actual displaced population from origin $o$ to destination $d$ ($\overrightarrow{OD}$) could be calculated using the following equation:

$$\overrightarrow{OD} = \frac{\overrightarrow{od}(O + D)}{(o + d)} \tag{4}$$

where:

- $\overrightarrow{od}$ is the ratio of the flow of mobile phone users from origin $o$ to destination $d$;
- $(o + d)$ represents the combined number of mobile phone users at $o$ and $d$; and
- $(O + D)$ represents the combined population at origin $O$ and destination $D$.

### 2.2.7. Validation Process

The estimated users' displacement matrix from anonymized CDRs were first compared with the post-disaster building damage assessment from remote sensing data. The post-disaster building damage assessment was carried out by United Nations Institute for Training and Research (UNITAR), and the results show the total number of identified buildings in each neighborhood, total damaged buildings, and the corresponding percentage of damaged buildings [42].

In addition, the estimated users' displacement matrix from anonymized CDRs was compared with the Displacement Tracking Matrix (DTM) from an individual questionnaire survey conducted by International Organization for Migration (IOM), in coordination with the Government of Mozambique through the National Institute for Disaster Risk Management (INGC). IOM's Displacement Tracking Matrix (DTM) is a system to track and monitor displacement and population mobility. It is designed to regularly and systematically capture, process, and disseminate information to provide a better understanding of the movements and evolving needs of displaced populations. DTM has been implemented

in Mozambique since 2013 with contextualized forms and tools for disaster and crisis responses in coordination with the INGC [43].

The evaluation consisted of comparing the arrivals in each administrative neighborhood from CDRs and from DTM. However, for Beira city, only data of 5 out of 26 localities were available online (IOM and INGC) [14]. Based on Deville et al. [38], a model to predict demand for disaster support in areas where there is no information available was proposed. In order to derive the model, it was first assumed that the relationship between the arrivals in each administrative neighborhood from CDRs ($A_{cdrs}$) and the arrivals in each administrative neighborhood from DTM ($A_{dtm}$) is represented by:

$$A_{dtm} = \alpha A_{cdrs}^{\beta} \tag{5}$$

where:

- $\alpha$ represents the scale ratio; and
- $\beta$ represents the superlinear effect of arrival form DTM ($A_{dtm}$) on the arrival from CDRs ($A_{cdrs}$).

This can be transformed into Equation (6):

$$Log(A_{dtm}) = Log(\alpha) + \beta Log(A_{cdrs}) \tag{6}$$

Parameters $\alpha$ and $\beta$ were estimated using a linear regression on training data to model the relation between arrival from DTM and arrival from CDRs in each administrative neighborhood. If we had had enough DTM, we would have used a standard cross-validation procedure; i.e., we would have divided the dataset into two groups, training (corresponding to 30%) and validation samples (corresponding to 70%). The first randomly selected set (corresponding to 30%) could have been used to train the model to derive coefficients $\alpha$ and $\beta$. The accuracy assessment statistics (correlation r and Root Mean Squared Error (RMSE)) could be calculated on the independent evaluation set consisting of the remaining 70% of administrative neighborhood. However, due to the limitation of DTM dataset (only five data points were available), based on the values of $Log(A_{dtm})$ and $Log(A_{cdrs})$ a standard linear regression model was fitted and correlation coefficient between the two variables was determined.

## 3. Data

This section describes the data that were used to test the proposed method, namely, anonymized mobile Call Detail Records.

### *Mobile Call Detail Records*

Mobile CDRs refer to the records that mobile phone operators make when a user (1) makes a call, (2) sends a short message, or (3) when an app uses the mobile Internet, and they are used for billing purposes [29,44]. The data used in this research were provided by Mozambique's Communications Regulatory Authority (INCM) and are from one of the biggest Mobile Network Operator (MNO) in Mozambique, holding around 30% of subscribers, and the observation period includes 32 days (from 6 to 30 March and 2 to 8 April 2019). Table 1 shows an example of mobile CDRs. Mobile CDRs data consist of an anonymized International Mobile Equipment Identifier (IMEI) of Caller and Callee, anonymized International Mobile Subscriber Identifier (IMSI) of Caller and Callee, start time of activity, duration of the call, Location Area Code (LAC) of Caller and Callee, cell-tower Identifier (CELL-ID) of Caller and Callee, activity type (Call, SMS, and Internet), and connection type (2G, 3G, or 4G).

**Table 1.** Example of an anonymized mobile CDR.

| Attribute | Value |
|---|---|
| IMEI-CALLER (anonymized) | 3bd78673f3084c4bcc564580c028b83367c5f8489dc5ba63a68afb3383f0d2ce |
| IMEI-CALLEE (anonymized) | 450325fa618ca1d370b69f54c5c1f485b05a2599273c94187ca3682582653211 |
| IMSI-CALLER (anonymized) | 1d68927da0e0681a887a0ac721af3c911aa71d965f1706d83c61bcc82d59e856 |
| IMSI-CALLEE (anonymized) | 674f6ebeb6d350e69326af7ca86335e7d55b7fe89731d2a135cb53eb2ba1b743 |
| START TIME OF ACTIVITY | 2019-03-09 21:39:42 |
| DURATION (seconds) | 320 |
| LAC-CALLER | 5800 |
| CELL-ID-CALLER | 694,715 |
| LAC-CALLEE | 620 |
| CELL-ID-CALLEE | 705,688 |
| ACITIVY-TYPE | CALL |
| CONNECTION-TYPE | 3G |

Figure 6 shows the total number of mobile phone activities and subscribers per day. The total activities per day represents the daily sum of all the calls, SMSs, and internet records made within the study area. The total IMSI per day represents the daily sum of unique mobile phone users who had at least one activity within the study area in a particular day. From this figure, it is clear that when the cyclone Idai made landfall at the port of Beira on 14 March 2019, the mobile phone operator was highly affected and registered a drastic drop on 15 March 2019 in terms of the total number of activities and the total number of IMSI. On 16 March 2019 and 1 April 2019, the mobile phone operator failed to record the activities either due to electricity failure or the MNO failed to record the data. Moreover, on 17 March 2019, there is almost zero activity, and IMSI registered by the mobile phone operator. However, from 18 March 2019 up to the end of the study period (8 April 2019), there is a gradual increase in the number of mobile phone activities and subscribers per day with a drop on 30 March and 2 April 2019.

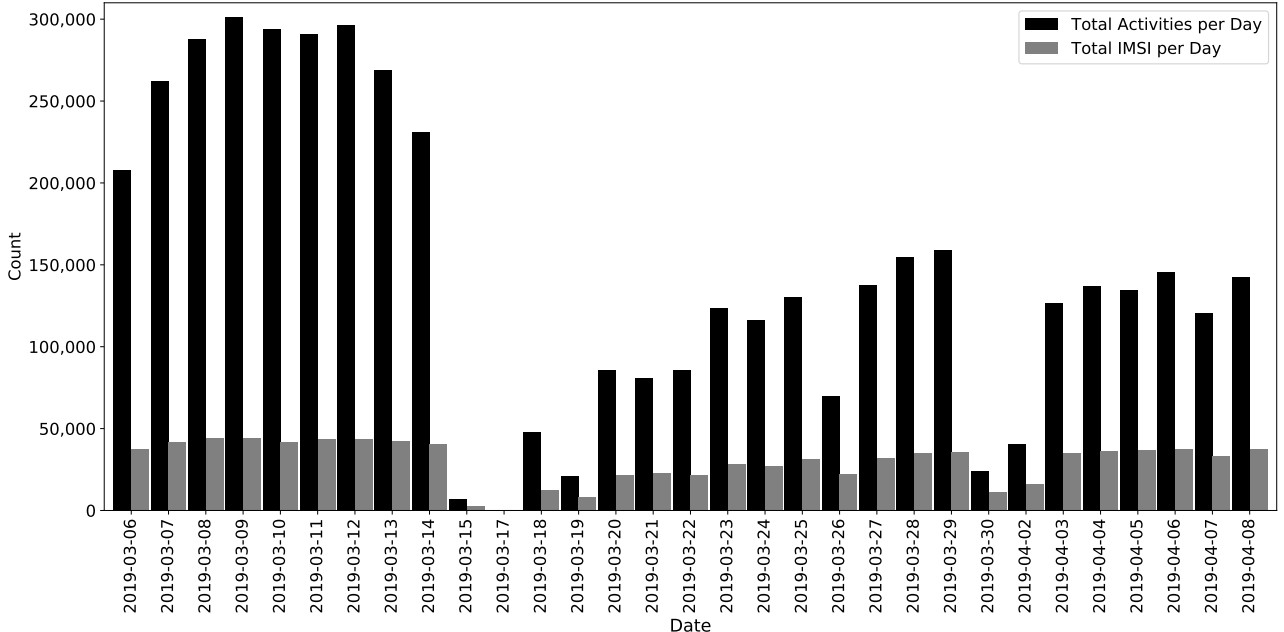

**Figure 6.** Total number of activities and IMSI per day in the study area.

In addition to the CDRs file, there is another file that contains spatial information (latitude and longitude), as well as the Cell-ID that is used to join CDRs with cell-tower data. The study area consists of 51 cell-towers (Base Transceiver Stations) irregularly distributed, i.e., densely distributed in the city center (high populated area) and sparsely distributed in suburban areas (less populated areas). Figure 7 shows the daily aggregated number of IMSI connected to each base station during the analysis period.

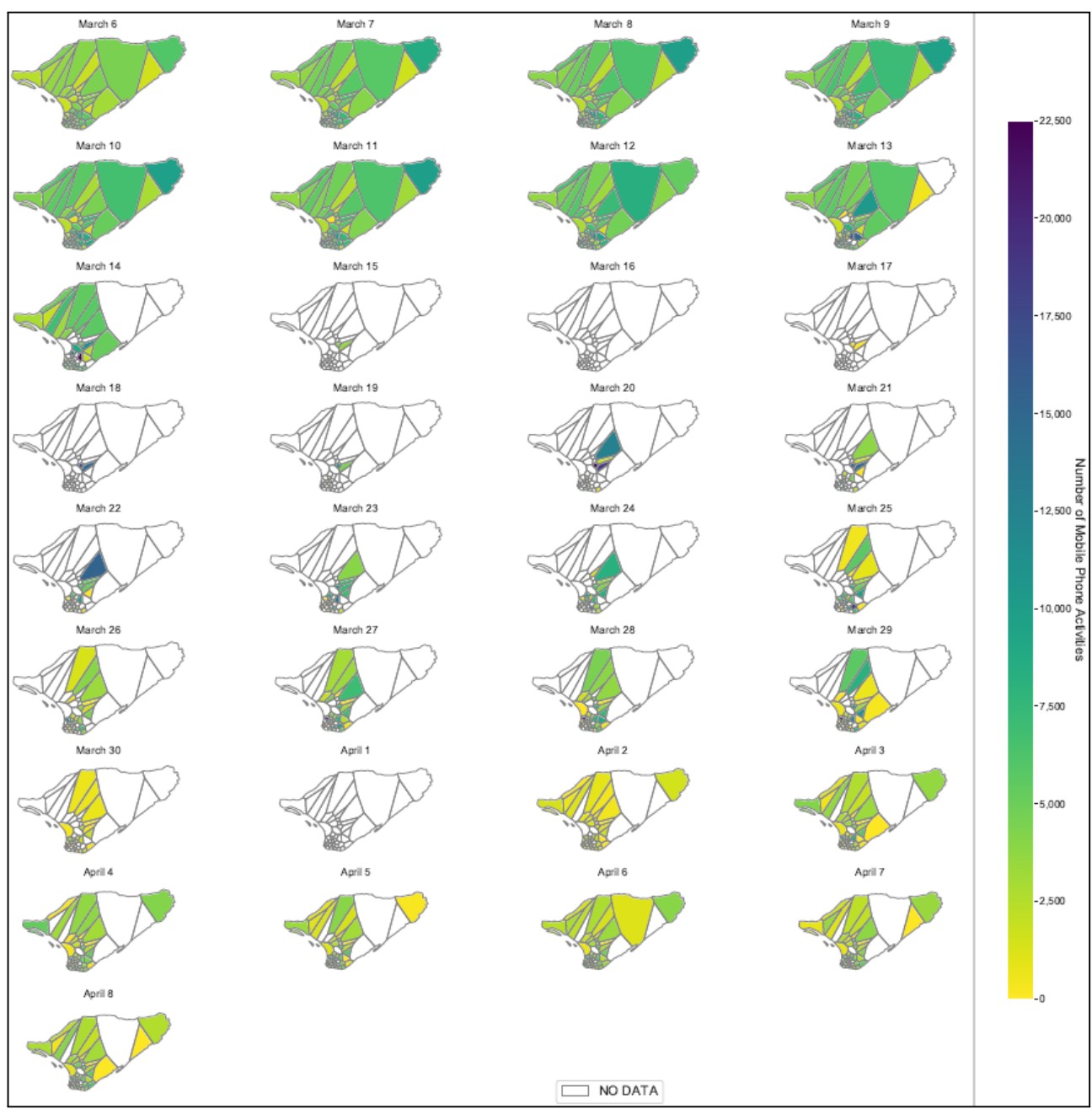

**Figure 7.** Daily Distribution of Mobile Phone Activities per Cell-Tower.

Figure 7 clearly shows that 14 March 2019 (the day that cyclone Idai landfall near Beira city) was the last day that the base transceiver stations registered the regular activities of mobile phone users. From 15 to 30 March 2019, some cell-towers did not register any mobile phone activity, represented by white color in each map and treated as NO DATA. This suggests that either the base stations were out of order (probably due to electricity failure or destruction) or all mobile phone users moved from the area covered by antennas to others. The choice of the cell-towers' operational problem was observed on 15 and 17 March 2019 when only two cell-towers were operational (registered some users' mobile phone activities). Nevertheless, it is possible to see the gradual recovery of the cell-towers after the disaster and the overload in some cell-towers whose neighbors were out of order. From 2 April 2019, over 78% of base stations were operational, but until 8 April 2019 (the last day of available data), there were still some cell-towers out of order. Even though Figure 6 suggests that from 18 to 29 March 2019 there is a gradual increase in the number of activities and subscribers registered by the mobile phone operator, Figure 7 shows that

many of the cell-towers during this period were out of order. Therefore, to reduce biases due to false displacement, in this study, the periods before and after the disaster were considered to be 6 to 14 March 2019 and 2 to 8 April 2019, respectively.

## 4. Experimental Results, Validation, Discussion

This section presents the main results of the experimental study, describes the validation process carried out to evaluate the effectiveness of the proposed method, and discusses the results.

### 4.1. Experimental Results

After estimating users' home locations (at cell-tower level) before and after the disaster, these data were aggregated to compute the population of mobile phone users in each cell-tower before and after the disaster. Figure 8 presents the aggregated population obtained by summing out the distinct users in each cell-tower before and after the disaster.

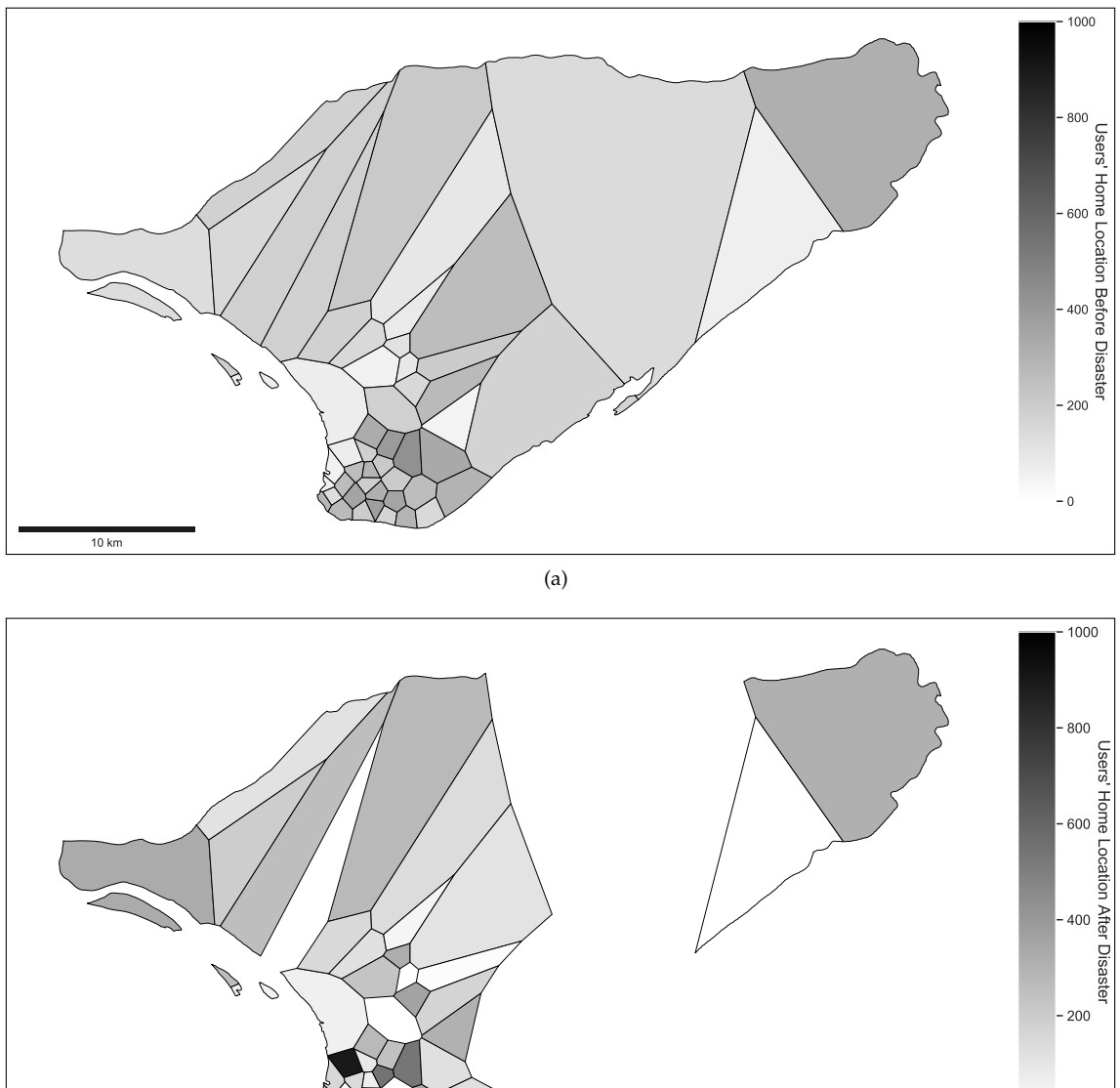

(a)

(b)

**Figure 8.** (**a**) Aggregated mobile phone users before the disaster and (**b**) aggregated mobile phone users after the disaster.

From Figure 8 it is possible to see the difference between the two aggregated mobile phone users' maps (before and after the disaster). The first visible difference is that Figure 8b comes with some blank areas (missing voronoi polygon), which represent the mobile phone cell-towers that were still out of order by the end of the study period (8 April 2019). Other visible differences can be seen in the central area of the city (area with dense cell-towers), where some cell-towers registered an increase in the mobile phone user population after disaster and others that decreased the number of anonymized subscribers. This was due to the fact that some of these areas were less affected, and therefore people moved-in, while other areas were highly affected, and people moved-out.

Based on the area of Voronoi polygon that falls within each neighborhood, a number of users who have to be assigned in each neighborhood administrative unit was calculated, and corresponding mobile phone users were randomly selected from mobile phone users who were found to be living in the Voronoi polygon before and after a disaster and assigned to the corresponding neighborhood. The aggregated assigned mobile phone users in each neighborhood before and after the disaster are presented in Figure 9.

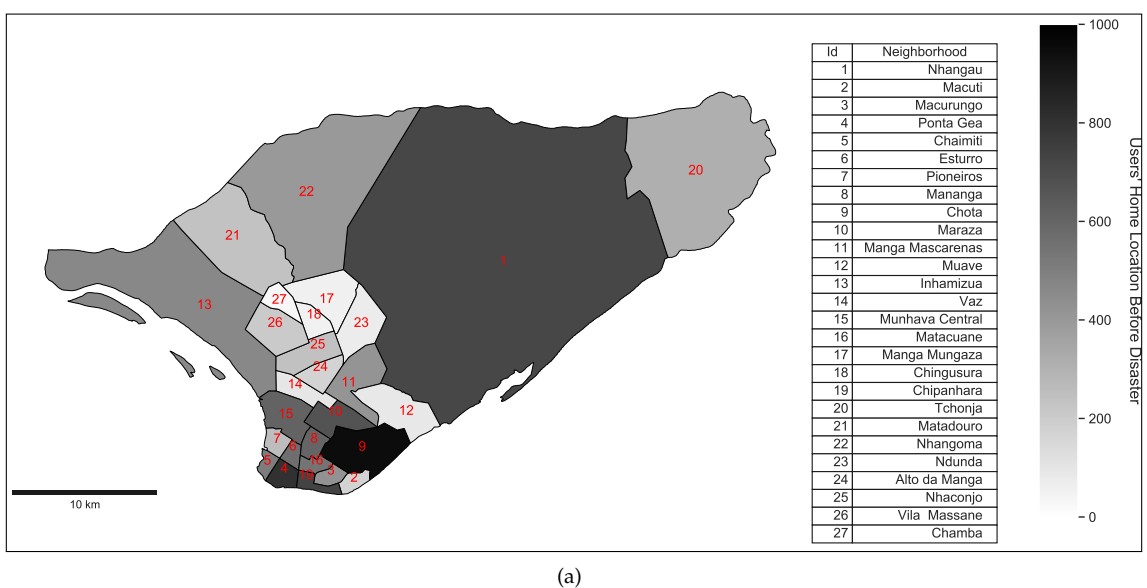

(a)

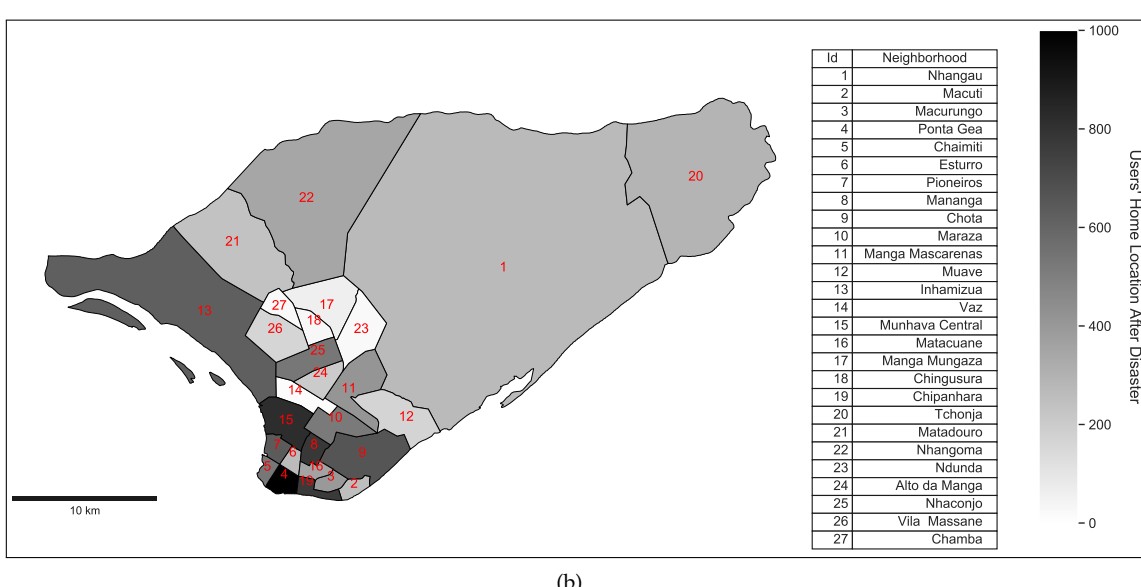

(b)

**Figure 9.** (**a**) Aggregated mobile phone users before the disaster at neighborhood level and (**b**) aggregated mobile phone users after the disaster at neighborhood level.

From Figure 9, some interesting differences can be seen. For example, in the Nhangau neighborhood (1), there is a drastic change in the number of mobile phone users who were assigned to this area. This is because the main cell-tower that covers this area was out of service for the entire period after the disaster considered in this study, as shown in Figure 8. However, since there are some other cell-towers that partially cover this neighborhood, it was possible to assign some mobile phone users to this neighborhood after the disaster. If the assignment of users was based on the coordinates of the cell-tower without considering the coverage area of each cell-tower as in Wilson et al. [25], the Nhangau neighborhood would not have any user after the disaster. This is an example that clearly shows how important it is to consider the coverage area of the cell-tower in the analysis.

The assigned users to the administrative neighborhoods were then aggregated based on their home location neighborhood ID before and after the disaster, and the mobile phone users' displacement matrix was derived. Figure 10 shows the mobile phone users displacement matrix.

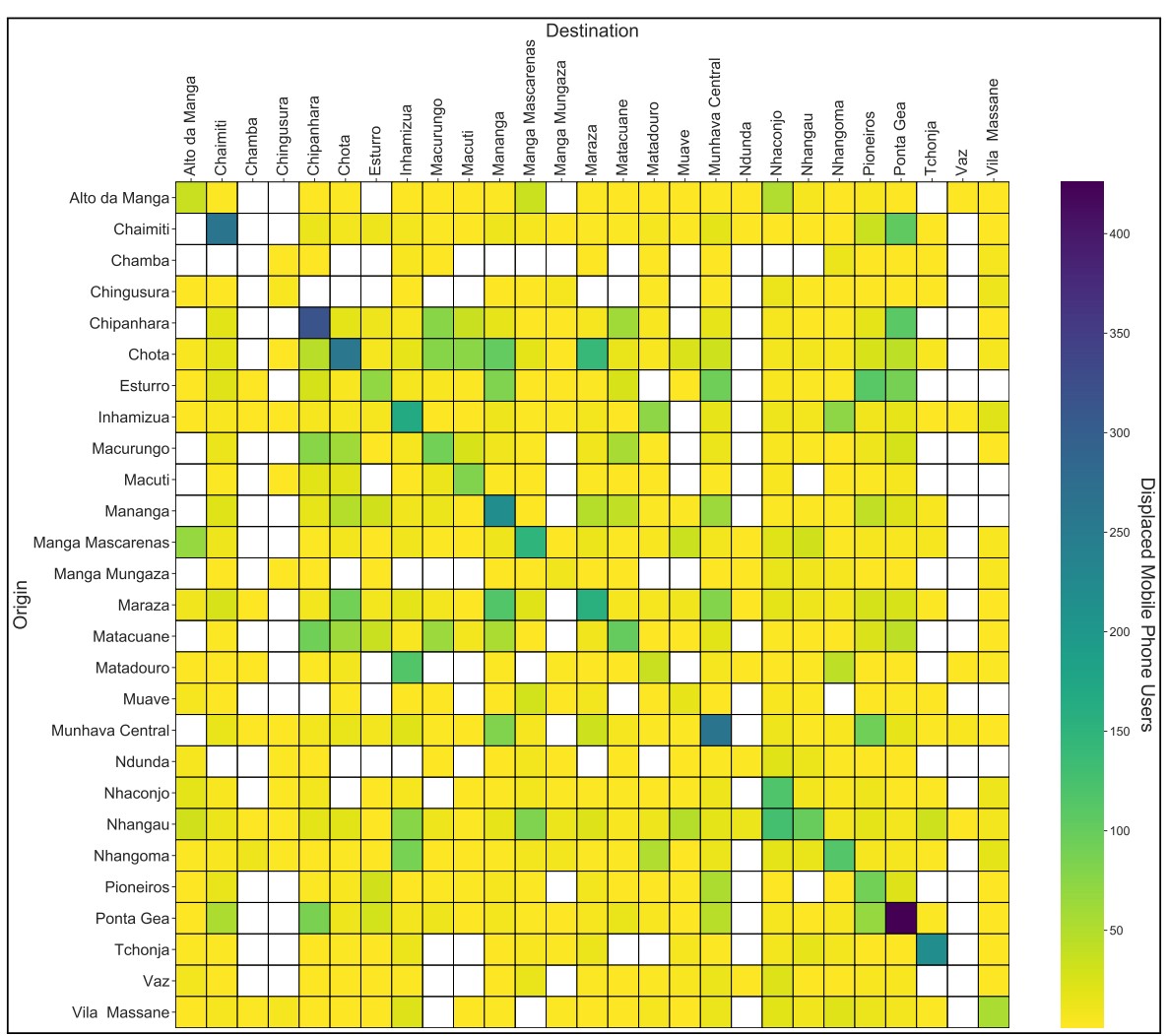

**Figure 10.** Mobile Phone Users Displacement Matrix.

In Figure 10, the diagonal northwest (NW)–southeast (SE) elements of the matrix represent the mobile phone users who were not displaced after disaster, i.e., the users who remained in their home neighborhood. The other elements in the matrix represent the mobile phone users' flow from origin to destination. Some neighborhoods registered a high number of mobile phone users who remained in their origin areas, namely Chaimite,

Chipanhara, Chota, Mananga, Munhava central, Ponta-Gea, and Tchonja. Among this neighborhoods, the highest number of non-displaced mobile phone users was registered in Ponta-Gea, which is justified by only 29% of damaged infrastructures in this area (the lowest damaged rate in entire Beira city) REACH [42]. Moreover, it is possible to see that few mobile phone users moved to Ndunda and Vaz neighborhoods. This is because these two neighborhoods were among the most damaged ones (both with damaged buildings over 80%) REACH [42].

The mobile phone users displacement matrix was then scaled up using the population data to obtain the actual flow among the origin-destination pairs. Figure 11 presents the scaled-up displacement matrix.

In Figure 11, the absence can be noted of three neighborhoods that were present in the mobile phone users displacement matrix (Figure 10), namely Chamba, Nhaconjo, and Tchonja. The reason for this is the unavailability of population data for these neighborhoods in the map where these data were extracted, available in [31].

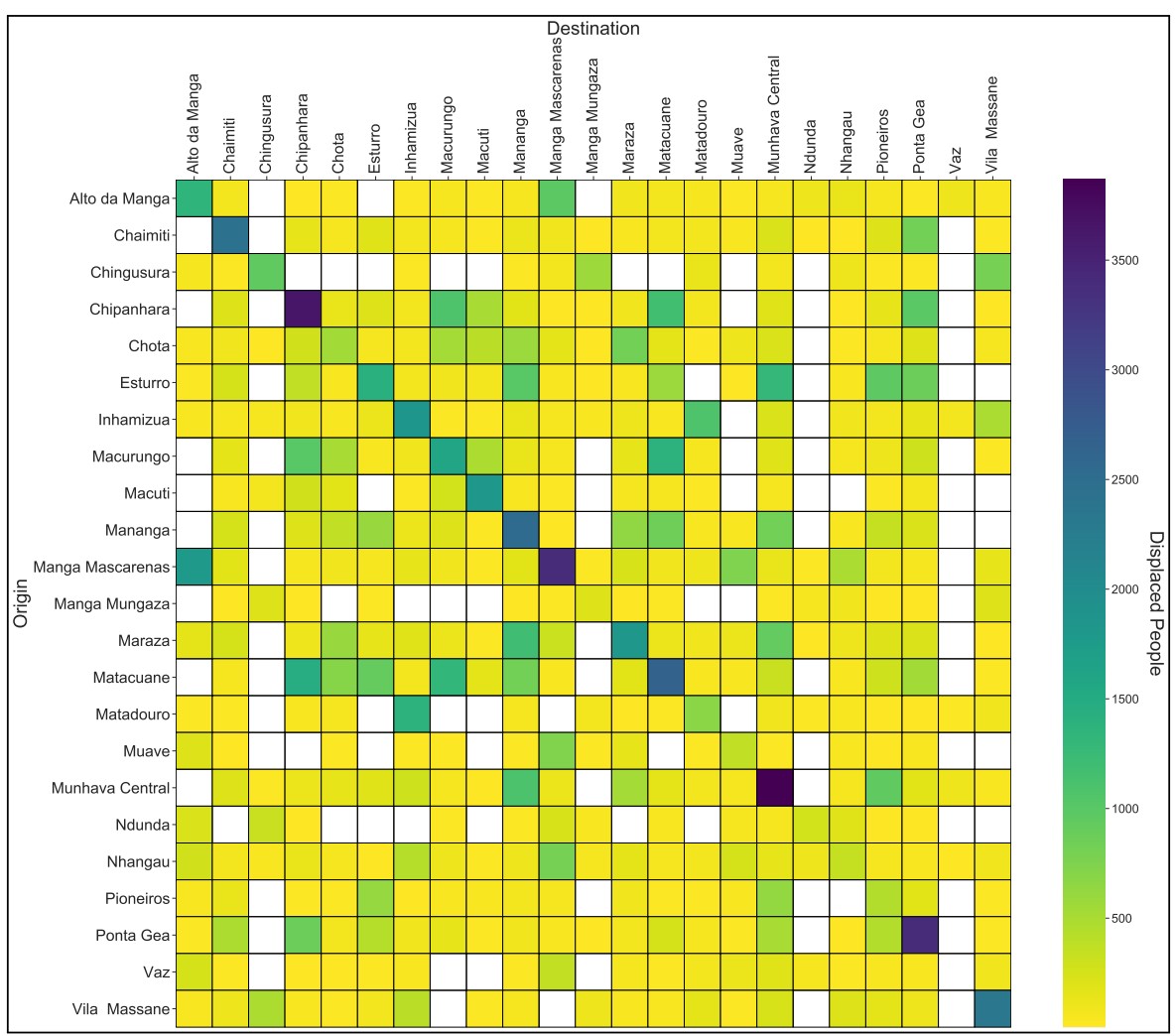

**Figure 11.** Actual population displacement matrix.

### 4.2. Validation of Results

The validation process consisted of comparing the *inflow, remain, and outflow* in each neighborhood and the percentage of damaged infrastructure derived from remote sensing techniques. Figure 12 shows the aggregated inflow, remain, and outflow in each neighborhood.

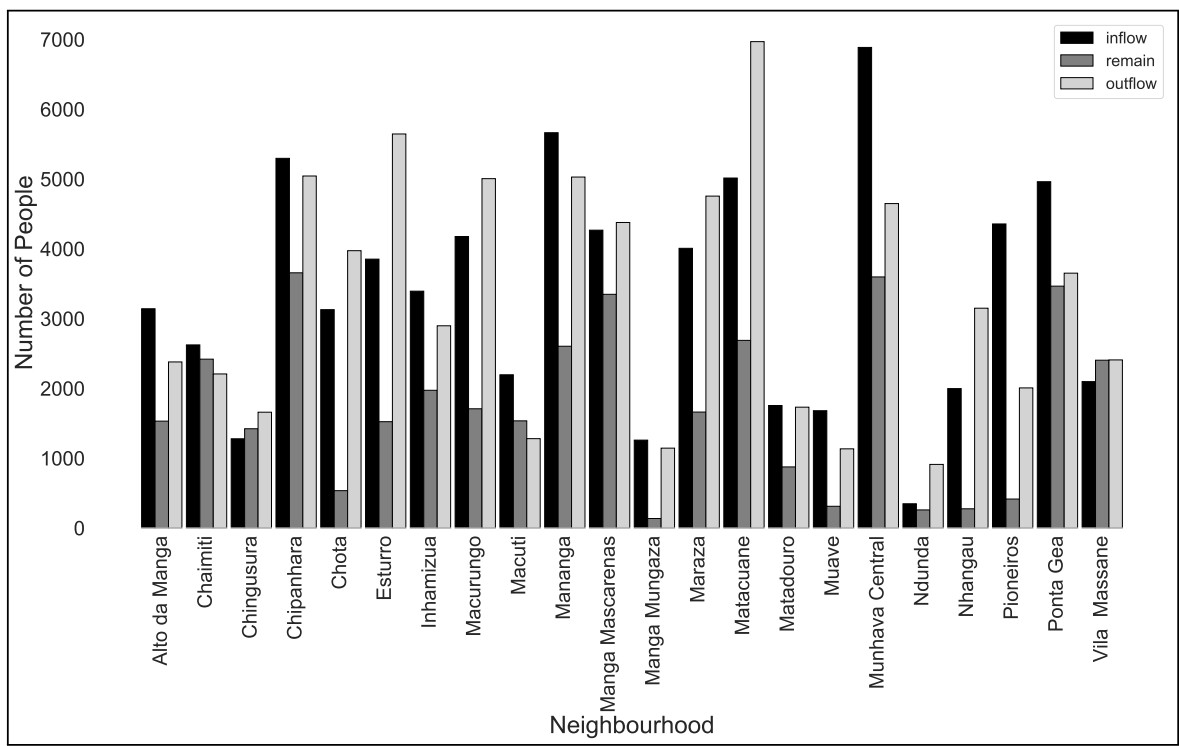

**Figure 12.** Aggregated inflow, remain and outflow from each neighborhood.

From Figure 12, it is clear that overall the number of people who remained in each neighborhood was less than the inflow (number of arrival in the neighborhood) and outflow (number of people who left the neighbor after the disaster). However, inflow and outflow vary from one neighborhood to another; i.e., in some cases, the inflow is greater than outflow (e.g., Ponta Gea, Alto data Manga, etc.), and in other cases, the inflow is less than outflow (e.g., Vaz, Esturo, Matacuane, etc.). These differences are supported by the idea that some neighborhoods were more highly affected than others REACH [42], as shown in Figure 13.

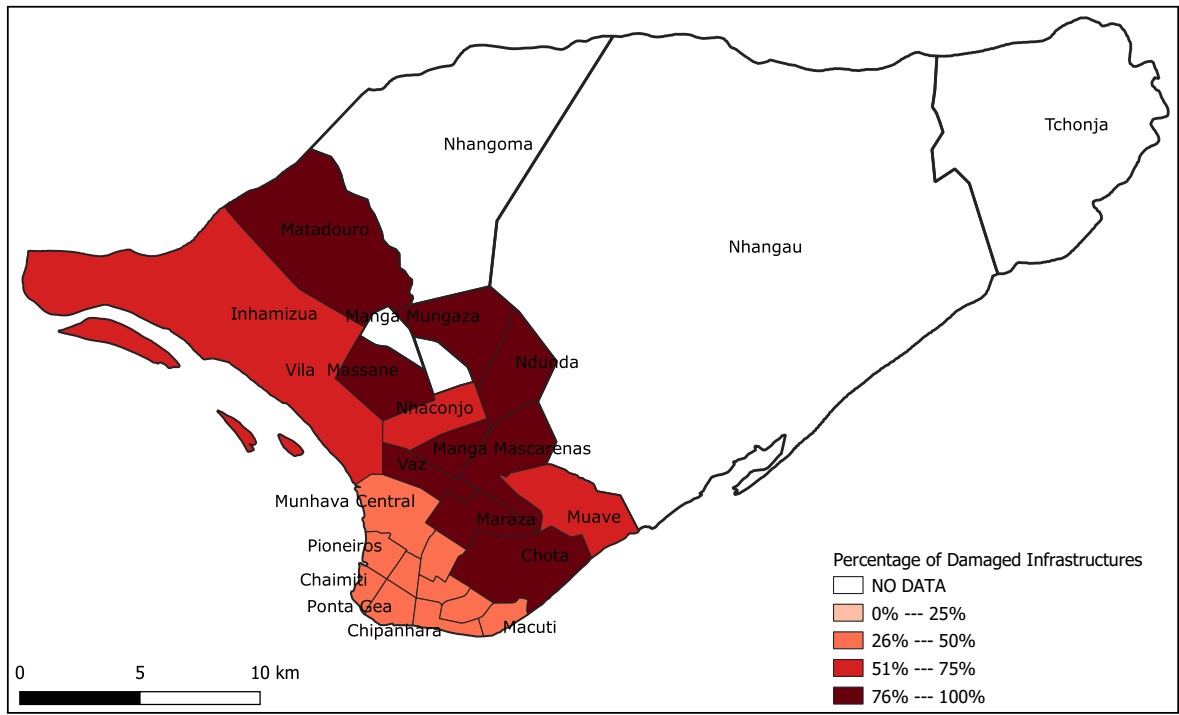

**Figure 13.** Percentage of damaged infrastructure per neighborhood.

For example, Ponta Gea and Munhava Central neighborhoods were less affected (only around 29% and 30% of infrastructures were damaged, respectively), and many people probably moved into this neighborhood. On the other hand, Chota, Vaz, and Maraza neighborhoods were highly affected (over 85% of infrastructure was damaged), and therefore, people might have moved out to other neighborhoods, which justifies the high outflow in these areas. In addition to this, in some neighborhoods where the difference between inflow and outflow is minimal, which means that the number of people that entered the area was relatively equal to the people that left it, the damaged infrastructures were between 40% and 50% (e.g., Macurungo and Chipangara). However, there are cases where the inflow was relatively greater than outflow, but still, the areas were highly affected (e.g., Mananga and Matadouro neighborhoods). Similarly, there are cases where the inflow is relatively less than the outflow, but still, the areas were less affected (e.g., Esturro neighborhood). The first case might have to do with the availability of shelters; i.e., even though the area was highly affected, there were enough shelters for the affected people coming from different areas. The second case might have to do with the flooded extent in these areas, i.e., the damage assessment data only show the percentage of damaged infrastructures and say nothing about the flooded areas. Therefore, even with fewer damages, the water level (e.g., in the Esturro neighborhood) might have forced people to leave their houses to other areas less affected.

Moreover, based on Equation (6) and the values of logarithms of arrivals in each administrative neighborhood from CDRs ($Log(A_{cdrs})$) and the arrivals in each administrative neighborhood from DTM $Log(A_{dtm})$, a standard linear regression model was fitted and correlation coefficient between the two variables was determined as shown in Figure 14.

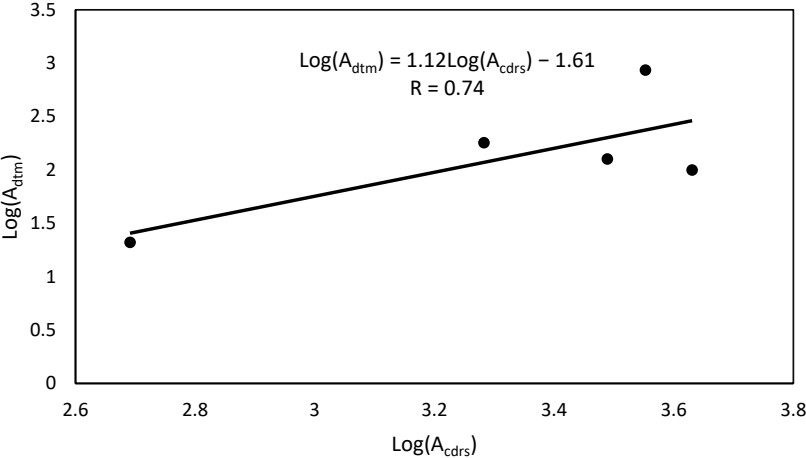

**Figure 14.** Comparison between the arrivals in each administrative neighborhood from CDRs ($Log(A_{cdrs})$) and the arrivals in each administrative neighborhood from DTM ($Log(A_{dtm})$).

Figure 14 reveals a promising correlation coefficient (over 70%) between the arrivals in each administrative neighborhood from DTM and the arrivals in each administrative neighborhood from CDRs. Therefore, if one can trust the model derived from the relationship between arrivals in each administrative neighborhood from CDRs and DTM, actionable knowledge can be extracted that can be used to predict the demand for disaster support in areas where no information is available.

However, some factors might have contributed negatively to the comparison results. The first factor has to do with the DTM sample data, which were limited (only data from five administrative neighborhood were available). The second factor is the data collection period used to estimate DTM (2 to 13 May 2019), which was a month after the period used to estimate the displacement matrix from CDRs (6 March to 8 April 2019). It is likely that when IOM and INGC conducted the survey, many people had returned to their origin administrative neighborhoods. Thirdly, the survey was conducted only in the official

identified shelters, but many people might have moved from their origin areas and likely were hosted by their families who were less affected by the disaster in other administrative neighborhoods. This argument is supported by the idea that when the cyclone hit Beira city, many official shelters were damaged, and the available ones were not enough to host everyone. The fourth factor has to do with coverage of CDRs used in this research; i.e., the estimated displacement matrix is relative to a particular mobile network company with its customer market share, under the assumption that the mobile network operator users are regularly distributed over the study area, which might not be true for some cases. Finally, CDRs only capture the mobility of people with mobile phones that were used before and after the disaster. However, the DTM captures the mobility of everyone who moved to shelters, i.e., the mobile phone users from the company that provided the data, mobile phone users from other companies, and non-mobile phone users including children, which might add bias in the comparison.

Even though the correlation is not very high, the CDRs data can still provide an idea about the people displaced by the cyclone. While the survey-based approach (by IOM and INGC) is time-consuming and demands human and financial resources to collect data to estimate the DTM, in some countries, mobile phone network companies provide anonymized CDRs for research institutions for social benefit with very low cost or no cost at all. If provided in a real-time fashion, CDRs can be used to derive the near-real-time displacement matrix, which can be used to support the disaster response teams in their activities.

### 4.3. Discussion

The study area consisted of 51 cell-towers which were distributed according to the population density, i.e., dense in highly populated areas (city center) and less concentrated in rural areas. Before the disaster, all the cell-towers registered the users' mobile phone activities regularly. However, when cyclone Idai struck Beira city, some cell-towers failed to register the users' mobile phone activities, probably due to electricity failure. This happened not only right after the disaster but was extended to 8 April 2019 (the last day of the analysis period).

Moreover, the study area consisted of 27 neighborhoods. Using the areal interpolation method proposed by Flowerdew et al. [28], it was possible to assign the mobile phone users in each cell-tower to the neighborhoods. This approach assumes that a cell-tower can be shared by different neighborhoods, which was not considered by Bengtsson et al. [23] and Wilson et al. [25], who assigned mobile phone users found to be living in each particular area covered by the cell-tower to a neighborhood where its coordinates fall into. As expected, all the neighborhoods in the study area were assigned mobile phone users, even in those cases where the main cell-tower was out of order for almost the entire period after the disaster (Nhangau neighborhood).

However, cell-towers' coverage areas were estimated using the approach proposed by Okabe et al. [26], which assumes that there is a homogenous distribution of mobile phone users within each Voronoi polygon. This is valid in highly populated areas with dense distribution of cell towers (usually within the city center). However, when it comes to those areas with less population density (rural areas), this approach is not realistic, because each antenna is designed to cover at maximum a particular area and due to the sparsity of cell towers, i.e., average distance between neighboring antennas over 7500 m in the rural environment, some areas in the study extent are not supposed to be covered [45]. In addition to this, the digital elevation model should be taken into consideration when estimating the coverage area of cell-towers [46].

The derived displacement matrix shows that a considerable number of people left their origin neighborhoods after the cyclone. This is supported by the damage infrastructure data; i.e., in the most affected areas, there is a high outflow and low inflow, and in the less affected area, there is high inflow and low outflow. However, in some cases, even though the area was highly affected (e.g., Mananga and Matadouro), the inflow was higher than the

outflow. This is probably due to the availability of shelters; i.e., even though the area was highly affected, there were enough shelters for the affected people coming from different areas. The opposite behavior was observed in Esturro neighborhood, where, even though the area was less affected, the inflow was relatively less than the outflow. This situation can be explained by the flood extent in this area; i.e., even with fewer damages, the water level in the area might have forced people to leave their houses to other less affected areas.

The comparison results between arrivals in each neighborhood extracted from CDRs and from DTM show an encouraging correlation coefficient (over 70%). Some factors can be pointed out as probable reasons for this encouraging but still low correlation coefficient. The first reason could be the fact that there is no overlap between the period of CDRs data collection and the survey conducted by IOM in collaboration with INGC used to derive DTM. Therefore, it is not possible to trace the DTM-related displacement during the time and compare it to the CDR-based results. In addition to that, in some neighborhoods located on the border of the study area, there is high probability that the arrivals presented on DTM are from outside the study, which may add bias in the comparison result since this was not considered in the CDRs-based analysis. The shared repository of the survey is available in DTM [47].

Since all the subscribers found to be living outside the study area before the disaster were ignored from the analysis, the method did not consider the inflow from response teams coming from different parts of Mozambique and other countries. However, by analysing the relationship between population distribution before the disaster derived from simulations analysis [48], infrastructure damages [49], and population behavior, a displacement hotspot would be identified before a disaster, which is valuable information for rapid disaster response. In addition to this, the method only focused on two periods, before the disaster (from 6 to 30 March), and after the disaster (from 2 to 8 April 2019), and the dynamics right after the disaster were not analyzed.

## 5. Conclusions and Future Work

In this research, an alternative method for displacement matrix after a severe disaster is proposed. The method uses mobile phone data, well known as CDRs, to estimate the spatial distribution of displaced people in each administrative neighborhood. The estimation starts by creating the Voronoi tesselation of the study area based on the coordinates of the cell-towers. Then, users' home location before and after a disaster is estimated. Based on the area of a Voronoi polygon that falls in each administrative neighborhood, mobile phone users found to live in this particular polygon are assigned to a different neighborhood using an areal interpolation method. Then, the users are labeled as displaced or non-displaced and then are aggregated based on the neighborhood before and after a disaster to derive the displacement matrix. However, the displacement matrix from CDRs only accounts for the mobile phone users' movement, not the population flow. Therefore, the estimated flow from CDRs was scaled up using the population data from the census survey. To evaluate the estimated displacement matrix, the derived inflow, remain, and outflow in each neighborhood was compared with the percentage of damaged infrastructures derived from remote sensing techniques. Furthermore, based on the relationship between inflow data (available only for five neighborhhods) from survey conducted by the International Organization of Migration (IOM) in collaboration with Mozambique's National Institute of Disaster Management (INGC) and inflow from CDRs, actionable knowledge was extracted, which can be used to predict the demand for disaster support in areas where there is no inflow information available.

However, while this study only focused on estimating the displacement only during two periods, before and after a disaster, it would be interesting to test the method for multiple periods after a disaster to capture the dynamics of the mobility of people after a disaster and hence determine the rate of return to the origin areas. In addition to that, by analyzing long period CDRs after a disaster, it would be possible to capture the resettlement areas of the displaced people. Furthermore, it would be interesting to evaluate

the dynamics of people immediately after the disaster, instead of considering only two periods as proposed in this paper. Therefore, a method to estimate the spatial distribution of displaced population immediately after disaster (when more cell-towers were out of order) needs to be developed. Moreover, while the approach presented in this paper focuses on estimating the number of displaced people after a disaster, it would be interesting to see correlation between the result of this study with the displacement hot-spot derived from the relationship between population distribution before disaster estimated using simulation models and the effects of damages on the population's behavior.

**Author Contributions:** Silvino Pedro Cumbane and Győző Gidófalvi contributed in the study conception and design of article. Silvino Pedro Cumbane also contributed in acquisition of all data, analysis and writing. In addition, Győző Gidófalvi provided comments for the revision of the paper. Both authors have read and agreed to the published version of the manuscript.

**Funding:** We would to acknowlodge Google for the data storage and processing platform through Google Cloud Platform credits (GCP credit coupon code: UTNV-07XD-T3RV-5XAQ).

**Institutional Review Board Statement:** Ethical review and approval were waived for this study, due to the use of anonymized mobile phone data.

**Informed Consent Statement:** People consent was waived due to the use of anonymized mobile phone subscribes' data in this study.

**Data Availability Statement:** The data is available and can be requested from INCM through submission of a research project.

**Acknowledgments:** We would like to thank the National Institute of Communications of Mozambique (INCM) for CDRs data provision used in this study. In addition we would like to thank the anonymous reviewers for their insightful comments, which significantly improved the paper.

**Conflicts of Interest:** The authors declare no conflict of interest.

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
