# Peer review of "Spatial Distribution of Displaced Population Estimated Using Mobile Phone Data to Support Disaster Response Activities"

_ijgi, doi:10.3390/ijgi10060421_

Round 1

Reviewer 1 Report

This paper is devoted to the use of mobile phone data in the scope of disaster management. This topic is interesting and current. I have already reviewed the previous version of the manuscript, and I think its quality has increased. I identified only the remaining minor issues. See below:

Introduction

  • The literature review was expanded compared to the first version of the manuscript, which I evaluate positively. I am now wondering if it would make sense to set aside a separate section on "Related Work."

Study Area and Methods

  • I do not consider the maps used in this section to be error-free (spatial resolution can be improved, etc.), but I think they illustrate the researched area better, now.
  • Even for Equation. 4, I would explain the variables using a bulleted list, whether it is the same in all cases.

Experimental Results, Validation and Discussion

  • Figure 6: Why is it not possible to visualize boundaries for all particular maps? Polygons, where the transmitter did not work, can also be displayed in a certain colour.

Author Response

Dear Reviewer 1,

Please find the revised version of manuscript ID: ijgi-1217689.

Reviewer 1 comments are valuable to improve the quality of this paper, and the paper has been revised according to Reviewer 1's comments. We believe that the paper is now of an acceptable standard after modification based on Reviewer 1's suggestions

The rest of this document is the details of our response to your comment:

Introduction

Comment 1: The literature review was expanded compared to the first version of the manuscript, which I evaluate positively. I am now wondering if it would make sense to set aside a separate section on "Related Work."

Response 1: Thank you very much for your comment. It would make sense to set aside a separate section on “Related Work”. However, we have decided to keep the structure of the previous version of the manuscript, i.e., we keep all the related work in the “Introduction” section.

Study Area and Methods

Comment 1: I do not consider the maps used in this section to be error-free (spatial resolution can be improved, etc.), but I think they illustrate the researched area better, now.

Response 1: Thank you very much for your comment. We have improved the spatial resolution of the maps. 

Comment 2: Even for Equation. 4, I would explain the variables using a bulleted list, whether it is the same in all cases.

Response 2: Thank you very much for your comment. We have updated this part in the revised manuscript. (Line 257 – 259)

Experimental Results, Validation and Discussion

Comment 1: Figure 6: Why is it not possible to visualize boundaries for all particular maps? Polygons, where the transmitter did not work, can also be displayed in a certain color.

Response 1: Thank you for your comment. The Figure has been updated to meet the reviewer’s 1 comment (see Figure 7).

Reviewer 2 Report

The paper has been slightly improved, but some requests were not taken into account by the authors. 

The main problem is that there is no authors’ reply to the previous reviewers' comment. This is not possible at this stage. This choice cannot effectively support me in the revision process, as well as it seems that the authors did not take into account all the reviewer’s suggestions. As a consequence, some lacks in the experiments are still pending.

DETAILED COMMENTS:

INTRODUCTION

  1. The term “panic” still exists in the main text (e.g. lines 68-73). Why? Which is its definition? I would like to encourage the author to avoid the term "panic", since many works (starting from Mawson's works) discussed the related phenomena by addressing specific behaviours affected by the surrounding environment and individuals. Please try to revise the related citations (i.e. 1) to better evidence which is de definition of panic you are using, or even if the word is not correct. Please also revise the abstract in this sense. “Panic” theories have been generally overcome since 1954 and Quarantelli’s perspectives.
  2. following point 1, people DO NOT act irrationally as natural response to physical danger. Line 32 is wrong. On the contrary, according to Mawson 2005, you can state that “People have an instinctive feeling as to the direction of safety. They tend to move away from danger and towards destinations perceived as safe” (see http://linkinghub.elsevier.com/retrieve/pii/S019897151100113X ), or that “This situation results in non-adaptive, competitive, and dangerous behaviours such as flight, pushing and trampling others to reach safety, and other violent conducts”  (see http://link.springer.com/article/10.1007/s11069-013-0688-9 ).
  3. “evacuation behavior of affected people”: affected by what? do you simply mean: “evacuation behaviours”?
  4. There was an improvement of the literature background as well as of the description of the aims. Nevertheless, I think you should better support the following sentence with some references: “Therefore, in this research, were adopted some proven techniques from different studies such as study 115 area tessellation, home location estimation, areal interpolation, and origin-destination estimation to derive 116 a method for the spatial distribution of displaced population after a disaster”. Furthermore, revise the structure of this sentence.
  5. Unsolved comment: why Mozambique is so important? what about the rest of the World?

METHODS:

  1. Figure 1: users’ home location before and after the disasters are based on the same methods. Why are you distinguishing them into two flows?
  2. Section 2.1: “Neighborhood is a district which is within Beira city.”: sentence with no meaning.
  3. “inhabitants density” should be “inhabitants’ density” (lines 224)
  4. “This cell-tower is shared among all the three neighbourhoods, B1, B2, and B3 and we assume that it has homogeneous areas in terms of inhabitants density”. Could you demonstrate that this statement is correct for your case study too?
  5. the questionnaire survey is a good way to validate the process, but maybe the questionnaire has to be available to the reader (also in an external source). If you will not provide the data, you have to explain why!
  1. Equation 5: what is the scale ratio? what does it mean? what is the superlinear effect? Maybe, a short discussion of the method by Deville et al. should be clearly provided in the methodology since the validation is a part of the work steps.
  1.  

DATA:

  1. pending ethical questions and data availability: What about some acknowledgement of the Mobile Phone Operator? Where and how data were and still are available? I think that if the authors will not provide further clear data on this issue, the paper cannot be published
  2. Figure 5: y-axis without the unit of measure

RESULTS

  1. Figure 6: white is NO data. Please point out this result!
  1. figure 13: why the number of points is lower than the neighbourhood numbers? This could affect the correlation. I think this is due to the poor DTM sample dimension. But please better explain this limitation when discussing the figure! This issue is not solved!

DISCUSSION

  1. DTM-CDRs correlations: the discussion is honest, but too many repetitions of concepts exist in section 4.2 (lines 404-423) and section 4.3 (lines 456-462). Please provide a unique discussion on the work limitations about this point. In addition, I can suggest you trace the DTM-related displacement during the time (if the survey included the day in which the survey was performed) and compare it to the CDRs-based results. On the contrary, please point out this additional limitation. Please also provide a shared repository of the survey.
  2. “the method did not take into consideration the inflow for example from the response 465 teams which was high to some extent.”: revise the English structure of this sentence
  3. subscribers living outside the study area before the disaster": the limitation is correctly remarked in the discussions section, but no connection with literature works are provided, since many methods exist to evaluate the number of people before the disaster, also according to a simulation-oriented approach, e.g. DOI: 10.1016/j.ijdrr.2018.12.010, DOI: 10.1016/j.culher.2020.12.007. You included some works on the temporal and spatial distribution of people in the introduction, but your efforts in the discussion section are still poor
  4. no discussion on the shape and different sizes of Voronoi polygons and their effects on the results is clearly provided.

CONCLUSION:

  1. In my previous comment, I pointed out that: “Conclusions are well addressed. I just invite the authors to discuss possible correlation or results with simulation models to be used before the event, as a support system in such a disaster. In addition, such a kind of data could be used to evaluate which "hot spot" in the city should be identified to limit the displacement because of the effects of damages on the population's behaviour”. This issue was not implemented. Why?

OTHER REMARKS: please accurately checked all the English sentences errors. The modifications to the papers seem to be not proofread.

Author Response

Dear Reviewer 2,

Please find the revised version of manuscript ID: ijgi-1217689.

Reviewer 2 comments are valuable to improve the quality of this paper, and the paper has been revised according to Reviewer 2's comments. We believe that the paper is now of an acceptable standard after modification based on Reviewer 2's suggestions

The rest of this document is the details of our response to your comment:

INTRODUCTION

Comment 1: The term “panic” still exists in the main text (e.g. lines 68-73). Why? Which is its definition? I would like to encourage the author to avoid the term "panic", since many works (starting from Mawson's works) discussed the related phenomena by addressing specific behaviours affected by the surrounding environment and individuals. Please try to revise the related citations (i.e. 1) to better evidence which is de definition of panic you are using, or even if the word is not correct. Please also revise the abstract in this sense. “Panic” theories have been generally overcome since 1954 and Quarantelli’s perspectives.

Response 1: Thank you very much for your comment. We have addressed this comment by replacing the term panic with “disaster risk perceived among homeless victims” as suggested at https://doi.org/10.1016/j.trb.2014.08.004 (Line 70-71)

Comment 2: following point 1, people DO NOT act irrationally as natural response to physical danger. Line 32 is wrong. On the contrary, according to Mawson 2005, you can state that “People have an instinctive feeling as to the direction of safety. They tend to move away from danger and towards destinations perceived as safe” (see http://linkinghub.elsevier.com/retrieve/pii/S019897151100113X ), or that “This situation results in non-adaptive, competitive, and dangerous behaviours such as flight, pushing and trampling others to reach safety, and other violent conducts”  (see http://link.springer.com/article/10.1007/s11069-013-0688-9 ).

Response 2: Thank you very much for your suggestion. We have updated the Line 32 to meet your suggestion as: “However, when a disaster occurs, even though there are predefined shelters such as schools, communal halls, libraries, and other buildings, in many cases, people have an instinctive feeling as to the direction of safety [1]. They tend to move away from danger and towards destinations perceived as safe [2].” (Line 31-31)

Comment 3: “evacuation behavior of affected people”: affected by what? do you simply mean: “evacuation behaviours”?

Response 3: Thank you for your comment. In the updated manuscript, this comment is addressed and term “affected people” has been removed. (Line 33)

Comment 4: There was an improvement of the literature background as well as of the description of the aims. Nevertheless, I think you should better support the following sentence with some references: “Therefore, in this research, were adopted some proven techniques from different studies such as study 115 area tessellation, home location estimation, areal interpolation, and origin-destination estimation to derive 116 a method for the spatial distribution of displaced population after a disaster”. Furthermore, revise the structure of this sentence.

Response 4: Thank you for your comment. The sentence was updated and references were added to support it as: “Therefore, in this research, were adopted some proven techniques from different studies such as study area tessellation [26], home location estimation [25,27], areal interpolation [28], and origin-destination estimation [27,29] to derive a method for the spatial distribution of displaced population after a disaster.” (Line 115 – 117)

Comment 5: Unsolved comment: why Mozambique is so important? what about the rest of the World?

Response 5: Thank you for your comment. We emphasized Mozambique since it our study area. However, we have decided to remove the sentence in the revised manuscript.

METHODS:

Comment 1: Figure 1: users’ home location before and after the disasters are based on the same methods. Why are you distinguishing them into two flows?

Response 1: Thank you for your comment. The method is the same but since the period considered after disaster (02 to 8 April 2019) is shorter, the condition that the frequency should be greater or equal to 2 is not considered in order to incorporate those users that have only one candidate home location after the disaster in the displacement matrix.

Comment 2: Section 2.1: “Neighborhood is a district which is within Beira city.”: sentence with no meaning.

Response 2: Thank you for your comment. The sentence was re-worded in the updated manuscript as : “Neighborhood is an immediate geographical area surrounding a family’s place of residence, bounded by physical features of the environment such as streets, rivers, train tracks, and political divisions [33]” (Line 138 – 140)

Comment 3: “inhabitants density” should be “inhabitants’ density” (lines 224)

Response 3: Thank you for your comment. This part has been revised in the updated manuscript. (Line 229)

Comment 4: “This cell-tower is shared among all the three neighbourhoods, B1, B2, and B3 and we assume that it has homogeneous areas in terms of inhabitants density”. Could you demonstrate that this statement is correct for your case study too?

Response 4: Thank you for your comment. Figure 5 presents an example of overlay between Voronoi and neighborhood administrative boundaries from study area. From Figure 5 it is clear that cell-tower 960 is shared among Maraza, Mananga, and Chota neighborhood.

Comment 5: the questionnaire survey is a good way to validate the process, but maybe the questionnaire has to be available to the reader (also in an external source). If you will not provide the data, you have to explain why?

Response 5: Thank for your comment. We did not conduct survey we used the results of the survey conducted by International Organization for Migration (IOM), in coordination with the Government of Mozambique through National Institute for Disaster Risk Management (INGC).  The shared repository of the survey is available in DTM [47].

Comment 6: Equation 5: what is the scale ratio? what does it mean? what is the superlinear effect? Maybe, a short discussion of the method by Deville et al. should be clearly provided in the methodology since the validation is a part of the work steps.

Response 6: Thank you for your comment. This part has been revised in the updated manuscript as following: “Parameters α and β were estimated by using a linear regression on training data to model the relation between arrival from DTM and arrival from CDRs in each administrative neighborhood.  If we had enough DTM, we would have used a standard cross-validation procedure i.e., we would have divided the dataset into two groups, training (corresponding to 30%) and validation samples (corresponding to 70%). The first randomly selected set (corresponding to 30%) could have been used to train the model to derive coefficients α and β. The Accuracy assessment statistics (correlation r and Root Mean Squared Error - RMSE) could have been calculated on the independent evaluation set consisting of the remaining 70% of administrative neighborhood. However, due to the limitation of DTM dataset (only 5 data points are available), based on the values of Log(Adtm) and Log(Acdrs) a standard linear regression model was fitted out and correlation coefficient between the two variables was determined.” (Line 285 – 294)

DATA:

Comment 1: pending ethical questions and data availability: What about some acknowledgement of the Mobile Phone Operator? Where and how data were and still are available? I think that if the authors will not provide further clear data on this issue, the paper cannot be published.

Response 1: Thank you very much for your comment. The data were provided by the National Institute of Communications of Mozambique (INCM). The data was produced by Movitel, one of the biggest Mobile Phone Operator in Mozambique. The data is available and can be requested from INCM through submission of a research project. (Line 300 – 303)

Comment 2: Figure 5: y-axis without the unit of measure

Response 2: Thank you for your comment. This has been revised in the updated manuscript (see Figure 6).

RESULTS

Comment 1: Figure 6: white is NO data. Please point out this result!

Response 1: Thank you for your comment. This part has been revised in the updated manuscript (see Figure 7).

Comment 2: figure 13: why the number of points is lower than the neighbourhood numbers? This could affect the correlation. I think this is due to the poor DTM sample dimension. But please better explain this limitation when discussing the figure! This issue is not solved!

Response 2: Thank you very much for your comment. This comment has been addressed in the updated manuscript by adding the following sentence: “The first factor has to do with the DTM sample data which was limited (only data from 5 administrative neighborhood were available).” (Line 424 – 426)

DISCUSSION

Comment 1: DTM-CDRs correlations: the discussion is honest, but too many repetitions of concepts exist in section 4.2 (lines 404-423) and section 4.3 (lines 456-462). Please provide a unique discussion on the work limitations about this point. In addition, I can suggest you trace the DTM-related displacement during the time (if the survey included the day in which the survey was performed) and compare it to the CDRs-based results. On the contrary, please point out this additional limitation. Please also provide a shared repository of the survey.

Response 1: Thank you very much for your comment. The section 4.3 (lines 456-462) was updated in the revised manuscript as: “The comparison results between arrivals in each neighborhood extracted from CDRs and from DTM show an encouraging correlation coefficient (over 70%).  Some factors can be pointed out as probable reasons for this encouraging but still low correlation coefficient.  The first reason could be the fact that there is no overlap between the period of CDRs data collection and the survey conducted by IOM in collaboration with INGC used to derive DTM. Therefore, it is not possible to trace the DTM-related displacement during the time and compare it to the CDRs-based results.  In addition to that, in some neighborhoods located in the border of the study area, there is high probability that the arrival presented on DTM are from outside the study are which add bias in the comparison result since this was not considered in the CDRs-based analysis. The shared repository of the survey is available in DTM [45].” (Line 481 – 489)

Comment 2: “the method did not take into consideration the inflow for example from the response 465 teams which was high to some extent.”: revise the English structure of this sentence

Response 2: Thank you for the comment. The sentence was revised in the updated manuscript as following: “Since all the subscribers found to be living outside the study area before the disaster were ignored from the analysis, the method did not consider the inflow from response teams coming from different parts of Mozambique and other countries.” (Line 490 – 492)

Comment 3: subscribers living outside the study area before the disaster": the limitation is correctly remarked in the discussions section, but no connection with literature works are provided, since many methods exist to evaluate the number of people before the disaster, also according to a simulation-oriented approach, e.g. DOI: 10.1016/j.ijdrr.2018.12.010, DOI: 10.1016/j.culher.2020.12.007. You included some works on the temporal and spatial distribution of people in the introduction, but your efforts in the discussion section are still poor

Response 3: Thank you for your comment. This part has been addressed in the revised manuscript as: “However, by analyzing the relationship between population distribution before disaster derived from simulations analysis [48], infrastructure damages [49], and population’s behaviour, displacement hot-spot would be identified before a disaster which is valuable information for rapid disaster response”. (Line 492 – 495)

Comment 4: no discussion on the shape and different sizes of Voronoi polygons and their effects on the results is clearly provided.

Response 4: Thank you for your comment. The discussion on this matter has been added in the revised manuscript as: “However, cell-towers' coverage areas were estimated using the approach proposed by Okabe et al. [44], which assumes that there is a homogenous distribution of mobile phone users within each Voronoi polygon. This is valid in highly populated areas with dense distribution of cell towers (usually within the city center). However, when it comes to those areas with less population density (rural areas), this approach is not realistic because, each antenna is designed to cover at maximum a particular area and due to the sparsity of cell towers i.e., average distance between neighboring antennas over 7500 m in rural environment, some areas in the study extent are not supposed to be covered [45]. In addition to this, digital elevation model should be taken into consideration when estimating the coverage area of cell-towers [46].” (Line 463 – 470)

CONCLUSION:

Comment 1: In my previous comment, I pointed out that: “Conclusions are well addressed. I just invite the authors to discuss possible correlation or results with simulation models to be used before the event, as a support system in such a disaster. In addition, such a kind of data could be used to evaluate which "hot spot" in the city should be identified to limit the displacement because of the effects of damages on the population's behaviour”. This issue was not implemented. Why?

Response 1: Thank you very much for your comment. This issue has been addressed in the updated manuscript as: “Moreover, while the approach presented in this paper focuses on estimating the number of displaced people after a disaster, it would be interesting to see correlation between the result of this study with the displacement hot-spot derived from the relationship between population distribution before disaster estimated using simulation models and the effects of damages on the population’s behaviour.” (Lines 525 – 529)

OTHER REMARKS: please accurately checked all the English sentences errors. The modifications to the papers seem to be not proofread.

Thank you for your comment. This issue has been revised in the updated manuscript.

Reviewer 3 Report

Dear Editor:

I think the authors has addressed all the reviewer's comments in the revised manuscript, and the quality of the paper has been improved. I recommend to accept it.

Author Response

Dear Reviewer 3,

Thank you very much for your response. We believe that the Editor will take into consideration your suggestion of accepting this manuscript for publication.

Best regards,

Authors

Round 2

Reviewer 2 Report

I would like to thanks their authors for their proper revision process. I just would like to encourage them to include the data availability statement (e.g. "The data is available and can be requested from INCM through submission of a research project.") as a footnote or into the acknowledgement section.

This manuscript is a resubmission of an earlier submission. The following is a list of the peer review reports and author responses from that submission.

Round 1

Reviewer 1 Report

This paper is devoted to the use of mobile phone data in the scope of disaster management. This topic is interesting and current. I identified partial shortcomings in the individual parts of the manuscript. I particularly recommend modifying (i.e., legends) and supplementing (i.e., background topographic data) all embedded maps.

See the following comments:

Introduction

  • The topicality of the topic can be documented by several studies that have dealt with this issue, for example, Li et al. (2019; https://ieeexplore.ieee.org/document/8788840), Kubicek et al. (2019; https://doi.org/10.1080/17538947.2018.1548654), Balistrocchi et al. (2020; https://nhess.copernicus.org/articles/20/3485/2020/), Wang et al. (2020; https://link.springer.com/article/10.1007/s10796-020-10057-w). Authors could add these studies to the literature review as they deal with the issue in other contexts. Data from mobile phones also have potential use in refining the estimate of the present population, which can differ significantly from official statistics.

Study Area and Data

  • I recommend adding the definition of "neighborhood”.
  • Figure 1: The map (in both map views) lacks graphic scales. It would be appropriate to modify the map and supplement the underlying topographic data to see, for example, the built-up areas, etc. Legible labels for” neighborhood” should be added to the detailed map view.
  • Row 123: “… from one of the biggest Mobile Phone Operators …” It would be appropriate to add information on how “big” the operator is – what market share it has. The operator’s name could also be given.
  • Figure 2:
    • There are two “gaps”. Missing data from 16 March 2019 are explained, but there is another gap on 1 April 2019. Even these missing data should be explained.
    • In legend is used the term “Total Activities per Day”. But what are the activities that are not included in the category "Total IMSI per Day".
    • From my point of view, both issues should be commented on in the text.
  • Figure 3: There are the same missing data (4–1) as in the previous figure.

Method

  • I think there would be a better title for this section, e.g., “Methodology” or "Methods".
  • Does the shape of Voronoi polygons affect the estimation?
  • All equations should be supplemented by a clear explanation of the variables used (in the form of bulleted lists).

Experimental Results and Validation

  • Figure 6: What do the missing Voronoi polygons mean?
  • Figure 7:
    • Labels are difficult to read.
    • How is it possible that the hot spot’s localisation differs from Figure 6?
  • Row 311: What does “NW-SE” mean? Are the individual “neighborhoods” located in Fig. 8 according to the orientation to the cardinal points?
  • Rows 353–363: Authors should move these paragraphs and the equations to the methodological section (3.6).

Conclusions and Future Work

  • I would expect a broader discussion of various aspects that may affect the results, such as:
    • spatially unequal market share of the given mobile operator,
    • different probabilities of occurrence of people in urban and undeveloped areas,
    • shape and different sizes of Voronoi polygons

Reviewer 2 Report

This work provides insights on an automatic system to trace post-disaster displacement in an urban scenario, at the wide-scale, according to the tracking of mobile phones activities. I think that this method is oriented towards the inhabitants (i.e. residents of the area) rather than on the visitors, including workers, and this fact should be clearly remarked. A preliminary verification through a case study is provided, although the validation is too rough and biased, as also pointed out by the authors. In general terms, this is an interesting paper. However, the literature discussion must be improved, as well as the methodological description, by providing a unique section on all the used methods (e.g. including figure 12 methods in Section 3.5). I think that the connection between the authors and the mobile phone operator should be disclosed in view of ensuring acknowledgement on how the input data were retrieved. Specific comments follow:

ABSTRACT:

  1. Please try to reduce the background of the wok and to give more details on the results also in a quantitative way.
  2. "The results show that CDRs derive spatial distribution of displaced populations with high coverage of people and in a timely fashion than traditional approaches." Please give better evidence of the results. What do you mean by "timely fashion"?

INTRODUCTION

  1. Please avoid to "copy and paste" part of the abstract into the introduction, since this is not useful to the reader (e.g. lines 25-26)
  2. I would like to encourage the author to avoid the term "panic", since many works (starting from Mawson's works) discussed the related phenomena by addressing specific behaviours affected by the surrounding environment and individuals. Please try to revise the related citations (i.e. 1) to better evidence which is de definition of panic you are using, or even if the word is not correct. Please also revise the abstract in this sense.
  3. "evacuation behavior of evacuees": this expression has no sense. You should better describe which kind of evacuees or using another expression.
  4. I think that some recent works on evacuation behaviours and relationships with the surrounding built environment should be included in the literature overview, to better define which are the context of the research (e.g. is it valid for all the disaster types?) and the behavioural issues which are considered by this research. I can suggest, e.g.: DOI: 10.1016/j.ssci.2019.104540 DOI: 10.1016/j.ssci.2020.104691
  5. The following requests 6 and 7 are similar. There are many example of methodologies in literature works, but there is not a comprehensive overview of the matter. Thus, the reader is not supported in understanding why the proposed method is innovative as well as which are the traditional methods for the same purpose, in a structured manner. I invite the authors to provide a short list of methodologies in the application context, e.g. by using a table or a bullet list, thus re-organizing the discussed methods into classes of methodologies.
  6. The discussion of traditional methods used for people displacement estimation should be improved. Just a couple of works are discussed. You should maybe focus on the main methods by avoiding long discussion on the specific case study, thus providing a list of approaches that can be used as traditional ones (e.g. by a bullet list).
  7. Lines 49-57: you should consider existing models tracing returning home behaviours to be more consistent with existing literature. Just as an example: http://dx.doi.org/10.1007/s11069-012-0175-8 ; http://dx.doi.org/10.1016/j.trb.2014.08.004 (this one include effects of "panic" behaviours)
  8. The research aim should be better clarified, in a short sentence. It is clear how the work contributes to the progress in the research field, but an "aim sentence" is needed.
  9. In addition to point 8, you point out that "some proven techniques from different studies": you should clarify which are these techniques in the introduction, in my opinion.
  10. "To the best of the authors’ knowledge, this is the first study in Mozambique": why Mozambique is so important? what about the rest of the World?
  11. Lines 99-103: the paper structure description is too detailed, since also methods are described. I think this part could be removed, and contents should be included in a methodological section (compare to point 1 of comments to the next section)

STUDY AREA AND DATA

  1. Before the case study description, a list of the phases is needed. You put a steps description in section 3, lines 158-165, but these steps should be linked also to the case study description, in my opinion. I think this is important since it allows the reader to replicate your work in other contexts. You can maybe use a workflow to this end (please compare with comment 1 to METHODS
  2. Figure 1 is not so relevant since just the neighbouring boundaries are described and, in addition, a metric scale is not reported. Which is the aim of the figure? Could you provide other data on the neighbourhood records (e.g. the number of inhabitants, risk levels, type of built environment...)?
  3. Section 2.2 actions could be clearer to the reader in view of a preliminary description of the methodological phases of the work.
  4. Figure 3. I think that a map with cell-towers should be provided to show which areas were more affected by the phenomena you are describing. You can also group the cell towers by homogeneous group for different areas. I know that some website can point out this kind of data (e.g. https://www.cellmapper.net/map?MCC=643&MNC=4&type=LTE&latitude=-19.839686211029147&longitude=34.88197199453251&zoom=11.090490963374288&showTowers=true&showTowerLabels=true&clusterEnabled=true&tilesEnabled=true&showOrphans=false&showNoFrequencyOnly=false&showFrequencyOnly=false&showBandwidthOnly=false&DateFilterType=None&showHex=false&showVerifiedOnly=false&showUnverifiedOnly=false&showLTECAOnly=false&showENDCOnly=false&showBand=0&showSectorColours=true&mapType=roadmap) As an alternative, I think that these data could be provided in an appendix, since most of the outcoming elements are also visible from Figure 2.

METHODS:

  1.  The list of phases is introduced here, but the complexity of section 3 methodologies is quite high, indeed. I think that a workflow could support the reader
  2. The main problem is that no clear correlation between cell-towers in FIgure 3 and the specific study area according to methods in section 3.1 can be effectively provided. Why there are 1000+ cells in Figure 31 and just 51 in line 167?
  3. "since all the subscribers found to be living outside the study area before the disaster are ignored from the analysis.": please remark such limitation in the discussions section, since this element affects the number of presences by non-residents in the area. This could be good for residential only areas, but not for the other ones, e.g. office areas, commercial areas and so on. Please point out this element also in the literature discussion, since many methods exist to evaluate the number of people before the disaster, also according to a simulation-oriented approach, e.g. DOI: 10.1016/j.ijdrr.2018.12.010, DOI: 10.1016/j.culher.2020.12.007
  4. This is just an issue concerning ethical questions and data availability: what about data on phone calls and users' positioning? Which database were used to derive the data? What about some acknowledgement of the Mobile Phone Operator? Where data are available?
  5. Assumptions for section 3.3-tower c6 should maybe refer to homogeneous areas in terms of inhabitants density. Is it correct?
  6. methods in section 3.5 should be better described, especially in relation to the scalar percentage of the market share of the mobile network operator you considered.
  7. Section 3.6: the questionnaire survey is a good way to validate the process, but maybe the questionnaire should be available to the reader (also in an external source)

EXPERIMENTAL RESULTS AND VALIDATION

  1. Units of measure in figure 6, figure 7 and figure 8 should be provided also for the coloured bars.
  2. You could assign a code to some of the areas where no people are present after the disasters, so as to better discuss the displacement/out of order reasons. For instance, risk maps or effects maps could be provided to better express the reasons for the displacement/out of order. This is a secondary issue, anyway.
  3. The Voronoi-base areas are really different from the neighbourhood areas, thus the outcoming discussion could be affected as well as the results. Maybe a list of the work limitations should be provided in this sense.
  4. Figure 8: please provide the boundaries of each column, since the matrix is really hard to be read.
  5. Lines 319-320: you are providing data on the damages/effects of the disaster. Could you provide a map of the effects, e.g. in terms of damaged buildings/infrastructure?
  6. the choice of the model in equation 5 and 6 is not clear. This aspect (and a short discussion of the method by Deville et al.) should be clearly provided in the methodology since the validation is a part of the work steps.
  7. figure 12: why the number of points is lower than the neighborhood numbers? This could affect the correlation.
  8. Lines 369-364: the limitations on the comparison results are well addressed, but there are some fears from my side about the significance of the comparison. In particular, the first factor (data collection period) can be a critical issue, thus affecting the correlation in figure 12. You should better explain the questionnaire structure to evidence if the difference in time could affect the results (e.g. questions were linked to the same period). 

CONCLUSIONS:

  1. Conclusions are well addressed. I just invite the authors to discuss possible correlation or results with simulation models to be used before the event, as a support system in such a disaster. In addition, such a kind of data could be used to evaluate which "hot spot" in the city should be identified to limit the displacement because of the effects of damages on the population's behaviou.

Reviewer 3 Report

The paper deals with spatial distribution of displaced population estimated
using Mobile CDRs to support disaster response. The paper appears interesting. In my opinion, there are still some problems worthy of discussion and improvement in this manuscript.

1) There are many studies on population migration and distribution using CDRs. The innovative contribution of this study should not only be to apply CDRs to the population spatial distribution of post disaster, expressed in lines 94-96. The authors are highly encouraged to review recent work on CDRs of the population spatial distribution, and to highlight the contribution of this paper in the part “Introduction”, "Disscussion" or "Conclusion".

2) It is strongly recommended to reference the research literature of CDRs in the field of population distribution in recent three years. Such as DOI: 10.3390/rs12162572.

3) The location of the study area shown in Figure 1 should have a compass, scale and geographic coordinates (latitude of longitude).

4) It should improve the description of the "Method" section, focus on how to get the geographical spatial distribution of population migration affected by disasters. Because if there is no disaster, there is also a spatial distribution of population migration.

5) Should a "Discussion" section be added?